# Low illumination fog noise image denoising method based on ACE-GPM

**Wuyi Li, Guanglu Zhou, Xingjian Wang** [ID]*

College of Computer and Control Engineering, Northeast Forestry University, Harbin, China

* jianxingwang@126.com

**Data Availability Statement:** All data files are available from the https://doi.org/10.6084/m9.figshare.25375222 All data files are available from the Github https://github.com/WuyiLi1999/Low_illumination_fog_noise_images.

## Abstract

The Perona-Malik (P-M) model exhibits deficiencies such as noise amplification, new noise introduction, and significant gradient effects when processing noisy images. To address these issues, this paper proposes an image-denoising algorithm, ACE-GPM, which integrates an Automatic Color Equalization (ACE) algorithm with a gradient-adjusted P-M model. Initially, the ACE algorithm is employed to enhance the contrast of low-light images obscured by fog and noise. Subsequently, the Otsu method, a technique to find the optimal threshold based on between-class variance, is applied for precise segmentation, enabling more accurate identification of different regions within the image. After that, distinct gradients enhance the image's foreground and background via an enhancement function that accentuates edge and detailed information. The denoising process is finalized by applying the gradient P-M model, employing a gradient descent approach to further emphasize image edges and details. Experimental evidence indicates that the proposed ACE-GPM algorithm not only elevates image contrast and eliminates noise more effectively than other denoising methods but also preserves image details and texture information, evidenced by an average increase of 0.42 in the information entropy value. Moreover, the proposed solution achieves these outcomes with reduced computational resource expenditures while maintaining high image quality.

## 1. Introduction

With the rapid development of information technology and computer vision, the application scope of digital image processing is expanding daily. Uncrewed Aerial Vehicles (UAVs) and cameras are widely used to obtain desired images in image acquisition work. However, the collected images are inevitably disturbed by noise, resulting in blurred images and low brightness [1]. These noises can be classified according to their characteristics, such as additive noise, shot noise, and multiplicative noise [1].

Additive noise [1] refers to adding noise to the image unrelated to the original image. This noise will be directly superimposed on the pixel values of the original image, resulting in changes in the brightness or color of the image. Additive noise may originate from electromagnetic interference of electronic devices, sensor noise, and many other factors. The Additive

**Funding:** This work was supported in part by Natural Science Foundation of Heilongjiang Province under Grant LH2020C048, and in part by the Harbin Science and Technology Innovation Talent Research Foundation under Grant 2017RAQXJ108." Funders played a vital role in supporting this research. Both funders were the main responsible persons for the project, contributing 10,0000 RMB each.

**Competing interests:** The authors have declared that no competing interests exist.

noise is usually randomly distributed, and standard additive noise models include Gaussian noise salt and pepper noise [1–6].

Shot noise [6] is the noise caused by the limitation or defect of the optical system, which will affect the image's clarity, contrast, and color accuracy. This noise is usually related to the image acquisition equipment, such as the camera lens, which may produce spots, color aberration, distortion, and other noise in the imaging process. The common shot noise is divided into aberration noise, chromatic aberration noise, defocus noise, halo noise, malformed noise, etc [2].

Multiplicative noise is a noise that affects the signal strength or energy of the image. Unlike additive noise, multiplicative noise can affect the overall brightness level of the image, reducing the contrast or loss of details in the image. Multiplicative noise is commonly seen in images under low light conditions, such as noise and graininess in images taken in the low light environment. Many types of multiplicative noise exist, including Rayleigh, Gamma, exponential, and fog [2].

Standard image noise denoising algorithms include filters based on spatial domain, wavelet threshold denoising based on wavelet domain [5, 6], image denoising based on Partial Differential Equation(PDE), image denoising based on Total variation (TV), and DnCNN denoising network model based on convolutional neural network. Filters based on the spatial domain are divided into mean filter, median filter, Gaussian filter, bilateral filter, and trilateral filter [2, 3].

In reference [2], a coupled Neural Network (PCNN) was combined with a regularized P-M equation to remove mixed noise in images. This method fully uses the biological heuristic characteristics of PCNN and introduces a threshold function to quickly locate the nonlinear noise information in the image. Then, the second-generation wavelet filter is used to suppress the positioning noise. Then, by regularizing the P-M model, the image's smoothness can be well controlled, and the edge detail information can be preserved while the residual noise information is removed. However, this method is only suitable for processing grayscale images and performs poorly when dealing with noise with unknown levels.

In reference [3], researchers used a color compensation method based on the designed attenuation matrix to improve the color distortion of the underwater image by compensating the red and green channels for the blue channel. In order to improve the local contrast of the image, they used a finite Bihistogram (DHIT) method based on the Rayleigh distribution. The proposed method can effectively enhance the details of the image and generate global and local contrast-enhanced images. Researchers have proposed a multi-scale unsharp mask strategy to improve the visual quality and sharpen the fused image further. These steps successfully improved the contrast of underwater and low-light images. However, it should be noted that adopting the DHIT method introduces a certain computational complexity, which may lead to the loss of some details.

In reference [4], in order to solve the problem of the high computational complexity of the Non-local Means (NLM) algorithm, researchers introduced moment invariant-based clustering and the Hidden Markov Model (HMM) for pre-classification. The proposed method exploits the Gaussian white noise dependence in the pixel wavelet transform domain to increase the number of candidate blocks for non-local means filtering, thereby improving the computational performance and denoising effect. However, this method contains multiple computational steps, such as clustering, wavelet transform, and Hidden Markov Model (HMM), and thus requires high computational resources when dealing with large images. In addition, this method is highly dependent on the data and can only suppress the additive white noise, which cannot adaptively deal with the noise of different scenes.

In reference [5], an image processing method based on minimum color loss and local adaptive contrast enhancement is proposed. The proposed method first combines the minimum loss principle with the maximum attenuation map to optimize the local color and details of the

input image. Then, the contrast is adjusted by adaptive local mean and variance, and the color balance is performed further in CLELAB color space to enhance the contrast and details of the image. Although the proposed method performs well in improving image contrast and brightness and enhancing detailed texture information, it still has certain limitations when dealing with images under low light conditions, and the effect needs to be improved.

In reference [6], an image processing method based on a semi-soft threshold model is proposed. Firstly, the proposed method can classify sharp edges of noisy images while avoiding excessive smoothing. Then, the self-adjusting generative adversarial network (GAN) model through the scoring machine is used to improve the quality assessment ability of the GAN model by adjusting the model parameters adaptively to generate higher-quality samples. This method can adapt to the noise characteristics of different images and perform effective image reconstruction. However, this model approach has some challenges regarding training complexity, hyperparameter sensitivity, and data requirements.

As mentioned above, to deal with the problem of significant computational complexity and easy loss of detailed information in image processing, this paper proposes a fog noise image processing method based on an automatic color equalization algorithm and gradient P-M model from the image gradient. The working block diagram of the proposed method is shown in Fig 1. The contributions of this paper are highlighted as follows:

1. This paper introduces an adaptive method, independent of large datasets, for processing low-light and fog-noise images with a single processor. Through extensive comparative experiments, the method's superior performance has been validated. It effectively addresses issues of insufficient contrast and brightness in low-light fog noise images and blurring of detail while ensuring high image quality and structure with a fast processing speed.

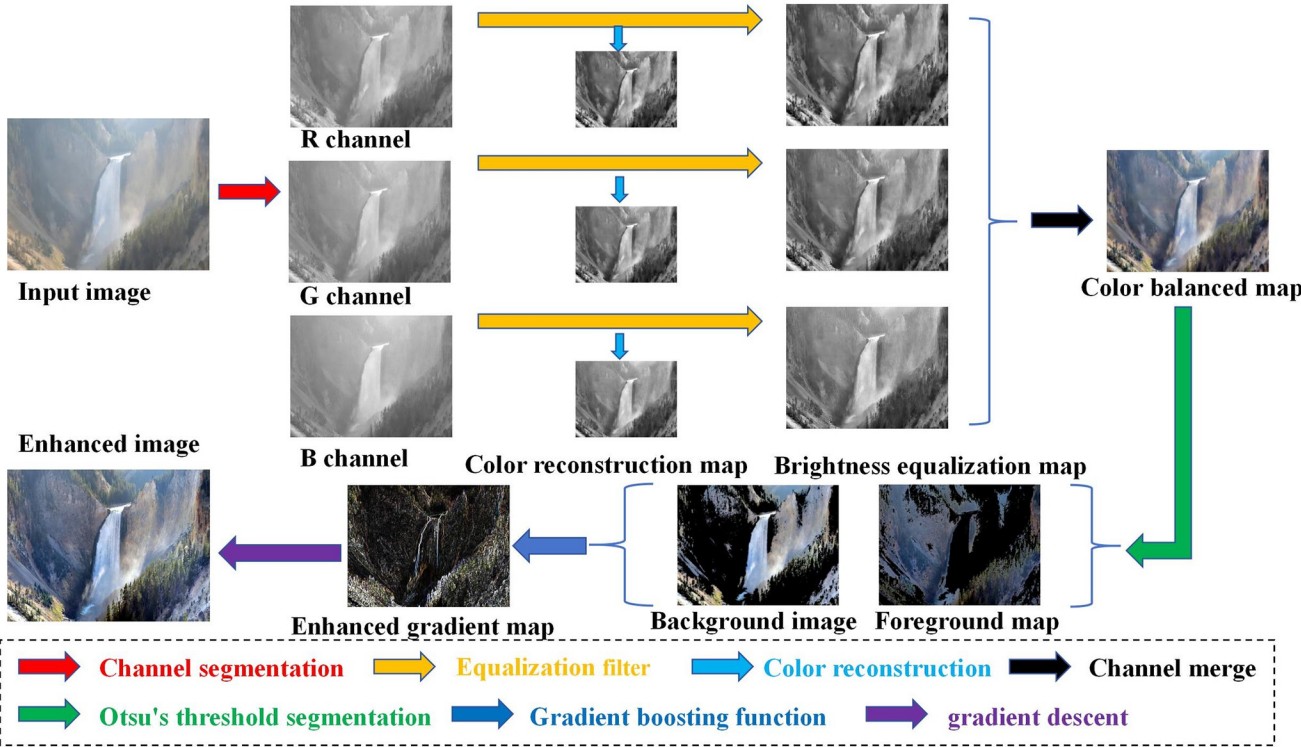

**Fig 1. The working block diagram of the proposed method.**

2. To mitigate the issue of noise amplification and loss of image detail during processing, the paper proposes a gradient-based P-M model method. The approach begins with threshold segmentation using the Otsu method to differentiate foreground from background. Subsequently, a gradient-based enhancement function is applied to accentuate the contrast between foreground and background, highlighting texture information and filling in gaps, thereby eliminating small-area noise.

The low-light fog noise image denoising algorithm proposed in this paper mainly involves the following three stages: preprocessing stage, enhancement stage, and post-processing stage.

1. **Preprocessing stage:** Firstly, the input image is segmented into the RGB three channels, and then the color space is reconstructed in each channel to obtain the corresponding color reconstruction map. Then, the equalization filter is used to equalize the reconstruction map of each channel, which can improve the brightness and contrast of the image while suppressing the noise. Finally, the processed image was output by channel merging.

2. **Enhancement stage:** The maximum between-class variance method(OTSU) performs threshold segmentation on the preprocessed image to determine an optimal threshold segmentation coefficient. Then, this threshold coefficient segments the image from the foreground and background. After that, different gradient-domain enhancements are performed on different segmentation regions to expand the difference between the foreground and background and highlight the image's edge texture information.

3. **Post-processing stage:** The gradient P-M model is established to denoise the gradient image in the enhancement stage, and the gradient descent method is used to solve the model and output the final result image. The proposed method can effectively suppress the noise and better retain the image's essential features to improve the image's quality.

## 2. Research methods

### 2.1 Automatic Color Equalization

Due to the influence of machines and environmental factors, the images acquired usually contain noise and are in low contrast. To solve these problems, Rizzi proposed the basic Automatic Color Equalization algorithm(ACE) based on Retinex and combined the theory of "gray world" and "white spot" [7, 8]. The proposed algorithm has remarkable results when dealing with high dynamic range images. By considering the position relationship between color and brightness, the adaptive filtering operation is carried out on local features to adjust the image brightness of local nonlinear characteristics [9]. In the brightness and contrast adjustment, the equalization filter dynamically adjusts the brightness to the appropriate range, thereby improving the image contrast. Low-light images usually have bright areas and shadow areas, so the brightness has a high dynamic range, so it is very suitable to use the automatic color equalization algorithm(ACE) to deal with low-light fog noise images [7]. The algorithm steps are shown as follows:

**Step 1:** Convert the input low-light color image into RGB color space.

**Step 2:** The color is reconstructed in RGB color space. The reconstruction function is shown in Eq (1):

$$R_c(p) = \sum_{j \in \text{Subject}, j \neq p} \frac{r(I_c(p) - I_c(j))}{d(p, j)} \tag{1}$$

Where $R_c(p)$ represents the reconstructed brightness of $p$ points in the RGB channel. $d(p, j)$ represents the distance between p and j points. $I_c(p) - I_c(j)$ is the luminance difference between points $p$ and $j$. $r(*)$ is the brightness expression function, and the function is shown in Eq (2):

$$r(t) = \begin{cases} 1, & t < -T \\ t/T, & -T \le t \le T \\ -1, & t > T \end{cases} \tag{2}$$

**Step 3:** The equalization filter is used to correct the color difference of the reconstructed RGB color space to improve the spatial brightness distribution of the image. The equalization filter formula is given in Eq (3):

$$O_c(p) = round[R_c(p) + q_c R_c(p)] \tag{3}$$

Where $q_c$ represents the slope of the line segment of $[(0, min[R_c(p)]), (255, max[R_c(p)])]$. $R_c(p)$ is the pixel value at point c after processing by the equalization filter. $round()$ is a function that converts floating-point data to an integer. Through simple linear expansion, the effect is close to human perception [7].

**Step 4:** The enhanced image is obtained by inverse transformation of the corrected image.

## 2.2 The Otsu's thresholding method

Otsu's thresholding algorithm is a widely used threshold segmentation method in image processing that can automatically determine the optimal threshold of the image. Based on the maximum between-class variance criterion, this method determines the optimal threshold by minimizing the between-class variance between background and foreground [10–12]. This method is robust, adaptive, and can effectively segment the image, so it is widely used in many fields.

When dealing with noisy images, the P-M model may introduce new noise, cause the loss of image detail information, and appear apparent gradient effects. To solve these problems, this paper uses Otsu's algorithm to segment the image and split it into two parts: foreground and background. The foreground part is mainly composed of pixels with higher gray values, which usually carry the detailed texture information of the image. The background mainly comprises low grey pixels, often containing the image's background, environment, and noise information. Through such segmentation processing, not only can the features of the image be better extracted and highlighted in the following enhancement processing, but the computational complexity of the image can also be reduced. The steps of Otsu's algorithm are as follows:

1. **Grayscale conversion:** Given that the input is a color image, Otsu's method is only suitable for processing grayscale images. To preserve the details and contrast information of the image more effectively, the weighted average method [13] is used to convert the input color image into a grayscale image.

2. **Compute the histogram of the grayscale image:** Calculate the histogram of the transformed image to obtain the number of pixels with gray value levels from 1 to m. The number of pixels of the gray value i is $n_i$.

3. **Compute probability distribution for gray levels:** Divide the number of pixels in each gray level by the total number of pixels(N) in the image to get each grey level's probability distribution. The formula for calculating the probability distribution of each gray level is shown in Eq (4):

$$p_i = \frac{n_i}{N} \tag{4}$$

4. **Calculate the between-class variance:** The between-class variance represents the degree of gray level difference between the background and foreground in the image. In the calculation process, the image's gray level is first divided into two categories: background and foreground for each possible threshold T [12]. Next, the weights and means of the two classes are calculated, and the within-class variance is obtained. Through this index, the image's distribution of different gray levels can be further understood, which provides strong support for subsequent processing and analysis. The within-class variance formula is given in Eq (5):

$$\sigma^2(T) = \omega_0(\mu_0 - \mu)^2 + \omega_1(\mu_1 - \mu)^2 = \omega_0\omega_1(\mu_1 - \mu_0)^2 = \frac{[\mu\omega_0 - \mu_0\omega_0]^2}{\omega_0(1 - \omega_0)} \tag{5}$$

Where $\omega_0$ is the weight of the foreground, $\omega_1$ is the weight of the background, $\mu_0$ is the mean of the foreground, $\mu_1$ is the mean of the background, and μ is the gray average of the whole image, $\omega_0 = \sum_{i=1}^{T} p_i$, $\omega_1 = \sum_{i=T+1}^{m} p_i$, $\mu_0 = \sum_{i=1}^{T} i*p_i/\omega_0$, $\mu_1 = \sum_{i=T+1}^{m} i*p_i/\omega_1$, $\mu = \omega_0\mu_0 + \omega_1\mu_1$.

5. **Find the best threshold T:** The best threshold T is found by iterating over all possible thresholds that maximize the variance between classes [13].

6. **Apply Threshold:** The selected best threshold is used to segment the image into background and foreground [12, 13].

In Fig 2, Images (a) and (d) are grayscale images processed by the ACE algorithm. Images (b) and (e) are the segmented background maps. Images (c) and (f) are the segmented

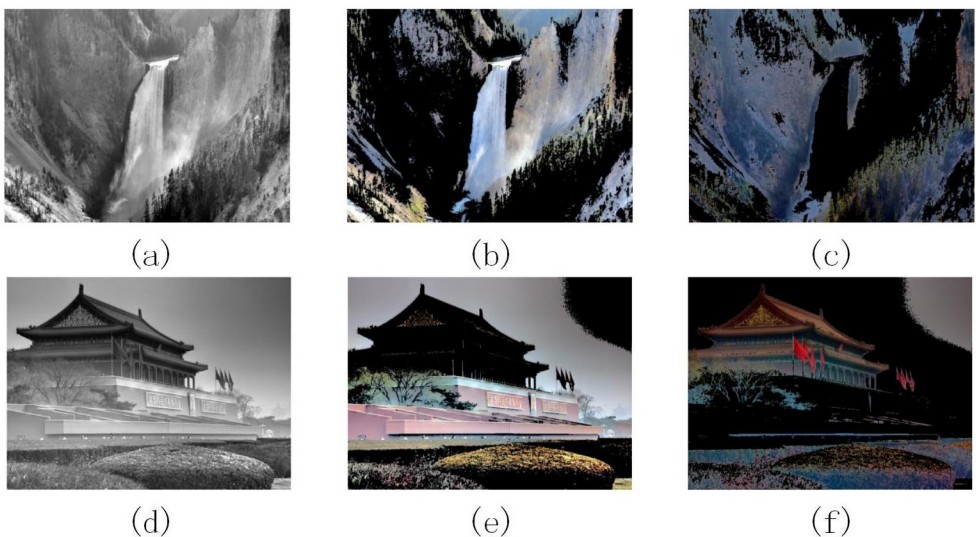

(a)　　　　　　　　(b)　　　　　　　　(c)

(d)　　　　　　　　(e)　　　　　　　　(f)

**Fig 2. Otsu's threshold segmentation diagram.**

foreground maps. According to the Otsu's algorithm, the image's optimal threshold segmentation coefficients T are 123 and 118, respectively.

## 2.3 Gradient boosting function

When dealing with noisy images, the original P-M model [2] may encounter problems, such as enhancing noise, introducing new noise, and producing significant gradient effects. To solve these problems, we propose to use the enhancement function to optimize the image gradient and reduce the occurrence of these adverse effects. Since the sharpness of the image edge is directly proportional to the gradient, the texture and edge detail information of the image can be reflected in different degrees according to the amplitude of the gradient field. Therefore, by adjusting the enhancement function, we can better improve the edge information of the image and thus deal with noisy images more effectively. The enhancement function of the gradient field is given in Eq (6):

$$f(|\nabla I|) = \begin{cases} 3 + \cos\left(\pi \dfrac{|\nabla I| - m}{T - m}\right) & |\nabla I| \leq T \\ 1.5 + 0.5\cos\left(\pi \dfrac{|\nabla I| - T}{M - T}\right) & |\nabla I| > T \end{cases} \tag{6}$$

Where $M$ is the maximum absolute value of gradient, that is, $M = \max(|\nabla I|)$. $m$ is the minimum absolute value of gradient, that is, $m = \min(|\nabla I|)$. $T$ is the determined segmentation threshold. $|\nabla I|$ is the absolute gradient of the image. So the enhanced gradient field is calculated as shown in Eq (7):

$$\nabla I = f(|\nabla I|)\nabla I \tag{7}$$

The gradient maps of the four images before and after enhancement are shown in Fig 3. Images (a) to (d) are the original gradient maps and images (e) to (h) are the corresponding enhanced gradient maps.

As shown in Fig 3, the image gradient processed by the enhancement function shows more significant features, and the image's edge contour also shows a more transparent and visible

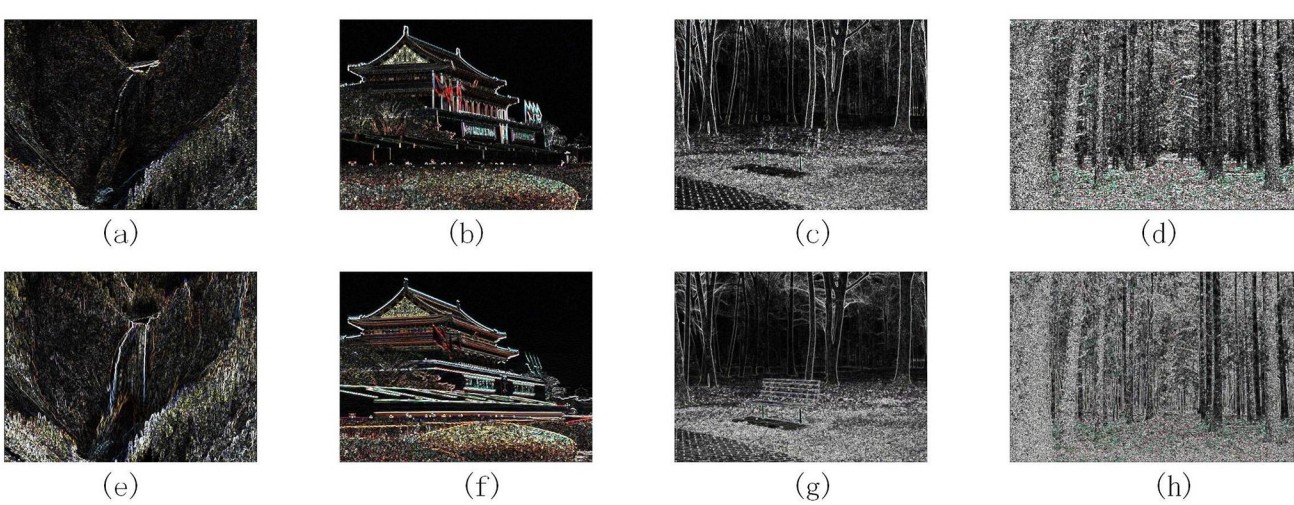

(a)  (b)  (c)  (d)

(e)  (f)  (g)  (h)

**Fig 3. Gradient maps of four images before and after enhancement.**

effect. In addition, applying the enhancement function can also improve the contrast of the image so that it presents a better visual effect and then highlights more detailed features.

This processing technique is significant for application scenarios such as image analysis, object detection, and recognition. Enhancing the image's edge contour and gradient information can improve the algorithm's accuracy and robustness to better deal with various challenges in practical scenes.

## 2.4 Gradient P-M model

The P-M anisotropic diffusion model based on partial differential equations was proposed by Perona and Malik in 1990 [2, 14–16] and applied to digital image processing. The model's primary function is to suppress noise and protect the edge texture features of the image. By applying this model, the influence of noise can be effectively reduced while retaining essential details and features in the image, which provides a better foundation for subsequent image analysis and processing. The anisotropic diffusion equation is shown in Eq (8):

$$\begin{cases} \dfrac{\partial I(x,y,t)}{\partial t} = div[c\|\nabla I\|\nabla I] \\ \\ I(x,y,0) = I_0(x,y) \end{cases} \tag{8}$$

Where $I(x, y, t)$ is the image function, $\nabla$ is the gradient operator, div denotes the divergence, $I_0(x, y)$ denotes the initial image with noise, and $c\|\nabla I\|$ is the diffusion coefficient equation. The P-M model is anisotropic diffusion [2, 14, 16]. It can protect the image edge information during diffusion.

The diffusion coefficient equation is given in Eq (9):

$$c\|\nabla \mathrm{I}\| = exp\left[-\frac{\|\nabla I\|}{k}\right)^{2}\right] \tag{9}$$

Where $\nabla I$ is the image gradient, and k is the electric heating coefficient.

Before applying the gradient descent method to solve the model, the gradient of the image should be first processed using the methods described in Parts B and C. This has several advantages:

1. **Enhanced contrast:** After implementing different levels of enhancement processing on the foreground and background, the contrast between the foreground and background can be strengthened, making the image more transparent and distinct.

2. **Removing noise:** The use of thresholding techniques and gradient enhancement can effectively assist the application of gradient descent in model parameter optimization, which is crucial for promoting the rapid convergence of the nonlinear diffusion process of PM models. In addition, the enhanced image can also improve the pertinence of the gradient-based PM model when dealing with noise in different regions, which can significantly improve the denoising efficiency. These improved measures provide strong support for the performance improvement of the model and the optimization of the practical application effect.

3. **Edge enhancement:** By using different gradient enhancement methods to differentiate the foreground and background, the contrast between the two can be increased, which helps the gradient-based PM model to identify and maintain these edges more accurately, reduce edge blur while denoising, and make the contour of the target object more visible.

According to the variational principle, the anisotropic diffusion Eq (8) is solved using gradient descent [17, 18]. Then, the minimized gradient descent flow function is given in Eq (10):

$$\frac{\partial I}{\partial t} = div\left[\rho'(|\nabla I|)\frac{\nabla I}{|\nabla I|}\right] \qquad (10)$$

Where $div$ is the divergence operator, $\nabla$ is the gradient operator, $t$ represents the iteration time, and $\rho$ is the diffusion coefficient function of the image at time $t$, which is used to control the anisotropic characteristics of the diffusion process.

Therefore, the finite difference method [19, 20]] is used as the discrete differential operator to solve Eq (10), and the discrete expression of image enhancement is obtained as shown in Eq (11):

$$I_{t+1} = I_t + \varphi(cN_{x,y}*\nabla_N(I_t) + cS_{x,y}*\nabla_S(I_t) + cE_{x,y}*\nabla_E(I_t) + cW_{x,y}*\nabla_W(I_t)) \qquad (11)$$

Where, $\nabla_N(I_t) = I_{x,y-1} - I_{x,y}$, $\nabla_S(I_t) = I_{x,y+1} - I_{x,y}$, $\nabla_E(I_t) = I_{x-1,y+1} - I_{x,y}$, $\nabla_W(I_t) = I_{x+1,y} - I_{x,y}$ is four divergence formulas, whose values are the partial derivatives of the current pixel in the directions of southeast, northwest, and northwest, respectively. $cN_{x,y} = \exp(-\|\nabla_N(I_t)\|^2/k^2)$, $cS_{x,y} = \exp(-\|\nabla_S(I_t)\|^2/k^2)$, $cE_{x,y} = \exp(-\|\nabla_E(I_t)\|^2/k^2)$, $cW_{x,y} = \exp(-\|\nabla_W(I_t)\|^2/k^2)$ is the thermal conductivity in the four directions of southeast, northwest, $I$ is the current image. $k$ is the thermal conductivity. $\varphi$ is step size.

## 3. Experiment

### 3.1 Datasets

In this study, we collected low-light fog noise images from domestic and foreign photography websites. We integrated these images into the experimental data set of the model in this paper. For detailed information about the image data sets, please refer to lot address: https://github.com/WuyiLi1999/Low_illumination_fog_noise_images.git.

### 3.2 Experiment setup

**3.2.1 Experimental environment.** The experimental environment of this paper: the system is a Windows 10 operating system, the processor is 2.50GHz Intel i5-7200U, and the memory size is 16G, the compiler is PyCharm2021.2.2, and the programming language environment is based on the 64-bit Python3.8 version.

**3.2.2 Baseline methods.** To verify the performance of the proposed denoising algorithm, several noisy color images under natural conditions were selected for visual and quantitative comparison analysis with eight different denoising algorithms. According to the selected literature, we briefly outline the method steps and give the corresponding evaluation information, which is shown in Table 1 below:

Ref. [22]: By finding the local minimum in the image histogram, the image is divided into multiple regions, and a specific range of gray levels is assigned to each region. Then, the dynamic histogram equalization of each region is carried out to achieve the denoising effect.

Ref. [21]: Firstly, the multi-scale Retinex theory decomposes the image into reflection and illumination components. Then, the brightness enhancement function (BEF) and the improved Adaptive Contrast enhancement function (IACE) were used to adjust the brightness of the illumination. Then, the Gaussian Laplacian pyramid fuses the two processed illumination maps. Finally, the enhanced illumination and reflection map were fused at multiple scales to obtain the final enhanced image.

**Table 1. Evaluation information for eight different denoising algorithms.**

| Method | Advantages | Disadvantages |
|---|---|---|
| Ref. [22] | The histogram equalization is adjusted according to the local characteristics of the image to better adapt to different regions and features and avoid over-saturation enhancement. | Artifacts or discontinuities are prone to appear at the boundaries.<br>New noise is introduced. |
| Ref. [21] | Increase contrast and brightness adaptively.<br>Good distinction between light and dark areas in the image. | The fusion operation will introduce noise, resulting in discontinuous boundaries and high computational complexity. |
| Ref. [23] | Effective smoothing of image noise.<br>The problem of boundary discontinuity is avoided.<br>The structure of the reconstructed image is preserved. | Details are easily lost. |
| Ref. [24] | Noise cancellation and brightness enhancement are performed while avoiding the generation of halos and artifacts.<br>An artificial bee colony is used to search for the optimal weight combination to improve the fusion effect. | It is computationally complex to find the optimal combination of weights.<br>The processing effect of different low-illumination images is not stable. |
| Ref. [25] | It avoids oversaturation of brightness and the introduction of new noise.<br>It has an excellent visual effect.<br>Ensure that the color is not distorted. | Each pixel's weighted least squares estimation takes a long time to compute and process.<br>The prospective fog treatment needs to be further analyzed and processed. |
| Ref. [7] | High computational efficiency.<br>Improve the contrast and brightness of the image. | Some details are lost.<br>It is prone to oversaturation. |
| Ref. [26] | Improves the contrast of the image.<br>Effective smoothing of image noise.<br>The structure of the reconstructed image is preserved. | High computational complexity.<br>The overall image is whitish.<br>Some details will be lost. |
| Ref. [27] | It improves the brightness and contrast of non-uniform illumination images.<br>Keep more details of highlighted areas. | The selection of weight parameters for processing low and high light areas in the light component is complex, and the calculation cost is high.<br>Part of the image structure is lost. |

Ref. [23]: Based on Retinex theory, the image is enhanced in HSI color space. A linear stretch is applied to the S component. The improved SSR algorithm processes the I component (bilateral filter instead of the original Gaussian filter). Finally, the Sigmoid function is used for color restoration.

Ref. [24]: Firstly, the illuminated component is extracted by relative total variation, and multi-scale Retinex obtains the reflected component. Then, the processing is enhanced with histogram equalization, bilateral gamma function correction, and bilateral filter. The copied original image is enhanced by histogram equalization using weighted guided filtering. Finally, the results of the two processes were weighted and fused by the Artificial Bee Colony (ABC) algorithm.

Ref. [25]: Adaptive low-light image enhancement algorithm based on weighted least squares method. Firstly, the original image is converted to LAB color space, and the L channel is processed by multi-scale detail enhancement and smoothing. then, the weighted least squares smoothing operator extracts the detailed information in the LAB channel and fuses it with the smooth features. Finally, multi-scale Retinex is used to enhance the brightness.

Ref. [7]: Automatic color equalization algorithm based on edge suppression mechanism and Gaussian distribution function.

Ref. [26]: Use the Particle Swarm Optimization (PSO) algorithm to select the optimal gamma value and use this gamma value for gamma correction.

Ref. [27]: Firstly, adaptive color balance is applied to achieve uniform color distribution. Secondly, the HSV color space is transformed, and the Gaussian function and multi-scale Retinex theory are used to extract the illumination and reflection components. Then, high-light and low-light regions are divided according to the pixel mean. Finally, the improved gamma correction algorithm was used to enhance the low light area, and the Weber-Fechner law was used to enhance the high light area to achieve the overall optimization.

**3.2.3 FOM merit evaluation index.** This paper uses the performance's quantitative evaluation indexes to evaluate the performance in the average luminance (*AL*), information entropy (*Entropy*), peak signal-to-noise ratio(*PSNR*), structural similarity(*SSIM*), mean square error of peak(*RMSE*), mean absolute error(*AME*), and mean square error of structural similarity (*SDME*). The mathematical formula of each evaluation index is as follows:

$$AL = (1/N) * \sum I(x, y) \tag{12}$$

$$Entropy = -\sum (p(x,y) * \log_2 (p(x,y))) \tag{13}$$

$$PSNR = 20 * \log_{10} (MAX^2/MSE) \tag{14}$$

$$SSIM = (2 * \mu_x * \mu_y + C_1) * (2 * \sigma_{xy} + C_2)/((\mu_x^2 + \mu_y^2 + C_1) * (\sigma_x^2 + \sigma_y^2 + C_2)) \tag{15}$$

$$RMSE = \sqrt{\frac{1}{N} \sum_{i=1}^{N} (I(x,y) - I'(x,y))^2} \tag{16}$$

$$AME = \frac{1}{N} \sum_{i=1}^{N} |I(x,y) - I'(x,y)| \tag{17}$$

$$SDME = \frac{1}{N} \sum_{i=1}^{N} (S(x,y) - S'(x,y))^2 \tag{18}$$

Where *N* is the total number of pixels, $I(x, y)$ is the pixel value of the image, $p(x, y)$ is the probability of occurrence of each pixel in the image, *MAX* is the maximum possible value of the pixel value, *MSE* is the mean square error, $\mu_x$ and $\mu_y$ are the average brightness of images x and y respectively, $\sigma_x$ and $\sigma_y$ are the standard deviation of brightness of images x and y respectively, $\sigma_{xy}$ is the brightness covariance of two images, $C_1$ and $C_2$ are constants (avoid the denominator being zero), $I'(x, y)$ represents the pixel value of the enhanced image, $S(x, y)$ is the structural information of the original image pixels, and $S'(x, y)$ is the structural information of the enhanced image pixels.

**3.2.4 Model parameter setting.** The P-M model has three key parameters: the number of iterations(T), the thermal conductivity(K), and the step size($\varphi$). In order to ensure the accuracy and effectiveness of the experimental results, the parameters set by the P-M model are shown in Table 2:

## 3.3 Experimental results and analysis

Figs 4–6 are images processed by each denoising algorithm with waterfall, bench, and Tian 'anmen Square as benchmarks, respectively. Tables 3–5 show the quantitative index data of images processed by each denoising algorithm using waterfall, bench, and Tian 'anmen Square as the benchmark images, respectively.

**Table 2. Model parameter configuration.**

| Parameter names | Parameter configuration |
|---|---|
| Number of iterations T | 20 |
| thermal conductivity K | 15 |
| Step size $\varphi$ | 0.15 |

According to the data in Figs 4–6, and Tables 3–5 show that the images processed by the method described in reference [22] has excellent performance in the average brightness index, so the color performance is relatively bright. However, this processing method may lead to over-enhancement, affecting the overall visual effect. In addition, some colors tend to have darker tones, and the problem of loss of details may occur, and discontinuities appear at the boundaries of the processed image.

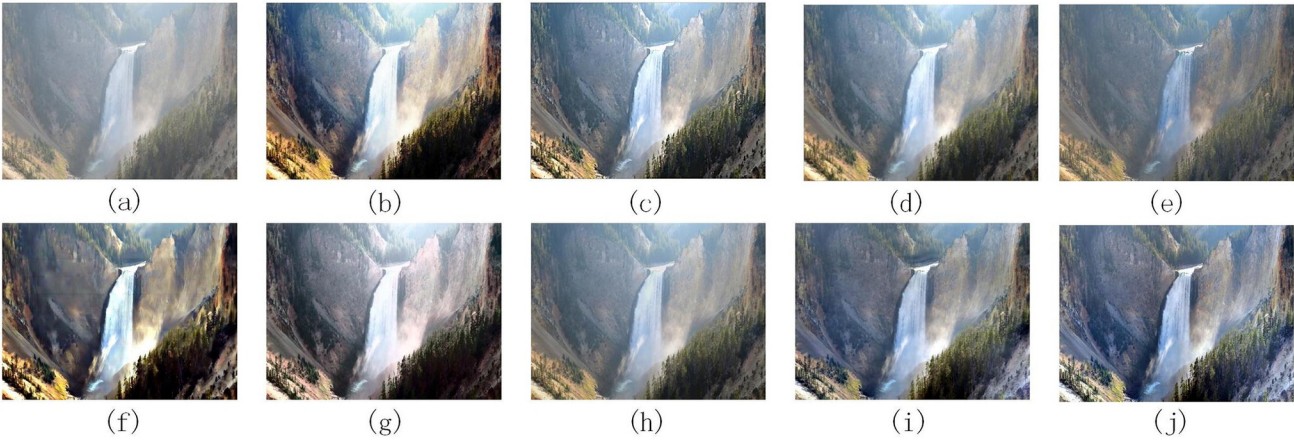

**Fig 4. The processing results of different denoising algorithms on waterfall images: (a) is the original image, (b) [22], (c) [21], (d) [23], (e) [24], (f) [25], (g) [7], (h) [26], (i) [27], (j) method is proposed in this paper.**

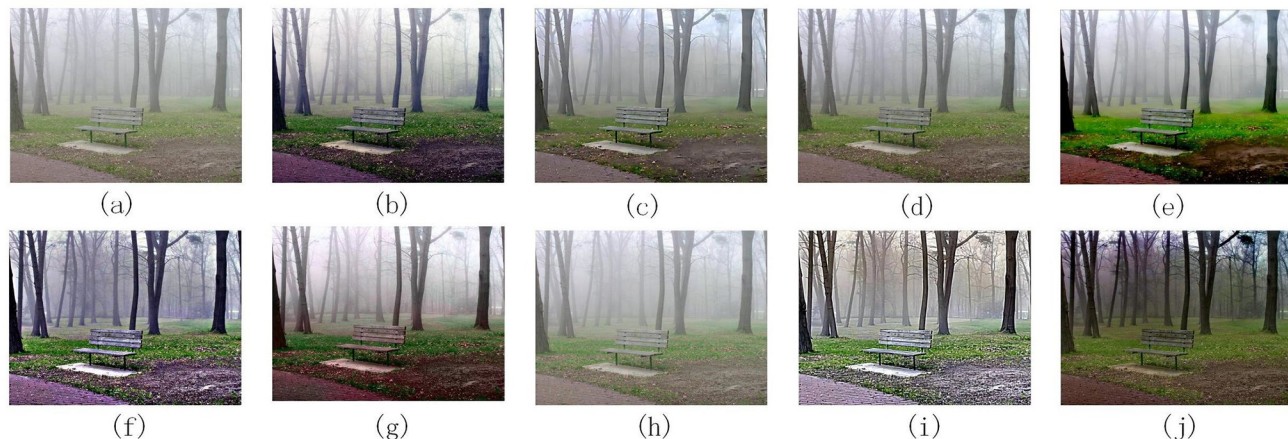

**Fig 5. The processing results of different denoising algorithms on bench images: (a) is the original image, (b) [22], (c) [21], (d) [23], (e) [24], (f) [25], (g) [7], (h) [26], (i) [27], (j) method is proposed in this paper.**

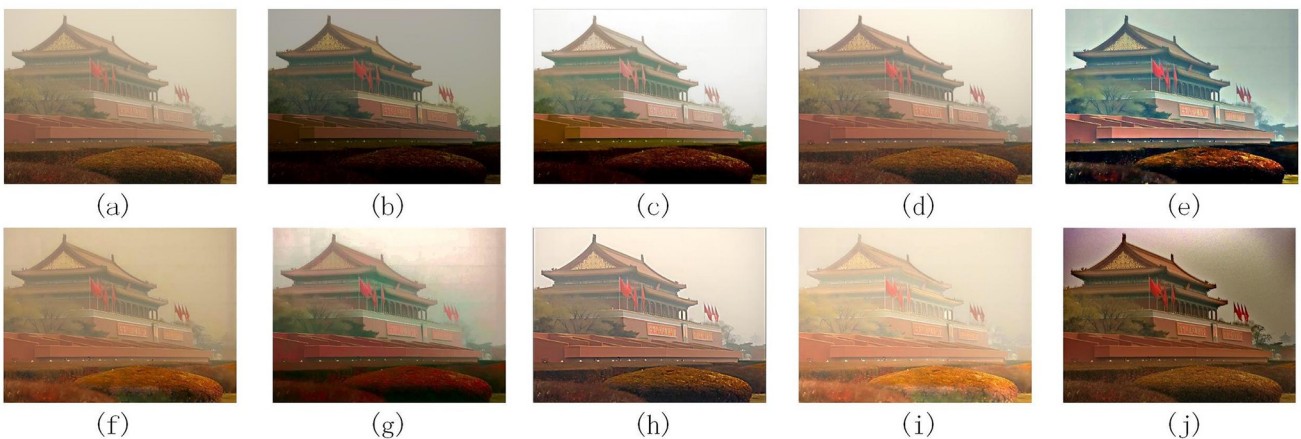

**Fig 6. The processing results of different denoising algorithms on Tian 'anmen Square images: (a) is the original image, (b) [22], (c) [21], (d) [23], (e) [24], (f) [25], (g) [7], (h) [26], (i) [27], (j) method is proposed in this paper.**

**Table 3. The evaluation index data of waterfall images processed by various denoising algorithms.**

| Method | AL | Entropy | PSNR | SSIM | RMSE | AME | SDME |
|---|---|---|---|---|---|---|---|
| Original | 90 | 6.96 | 20 | 1 | | | |
| Ref. [22] | 529 | 7.79 | 27.71 | 0.535 | 10.45 | 113.70 | 123.47 |
| Ref. [21] | 780 | 7.41 | 28.19 | 0.633 | 9.95 | 76.62 | 122.53 |
| Ref. [23] | 102 | 7.18 | 27.63 | 0.938 | 9.85 | 72.37 | 123.01 |
| Ref. [24] | 193 | 6.76 | 27.41 | 0.813 | 10.92 | 37.50 | 125.41 |
| Ref. [25] | 910 | 7.80 | 27.57 | 0.591 | 10.65 | 76.86 | 122.36 |
| Ref. [7] | 551 | 7.97 | 27.69 | 0.576 | 10.61 | 106.44 | 123.47 |
| Ref. [26] | 385 | 7.49 | 27.68 | 0.789 | 10.15 | 37.35 | 123.51 |
| Ref. [27] | 769 | 7.54 | 27.83 | 0.576 | 10.41 | 62.51 | 123.81 |
| **Ours** | **376** | **7.77** | **27.78** | **0.841** | **10.39** | **87.11** | **123.43** |

**Table 4. The evaluation index data of bench images processed by various denoising algorithms.**

| Method | AL | Entropy | PSNR | SSIM | RMSE | AME | SDME |
|---|---|---|---|---|---|---|---|
| Original | 1203 | 7.26 | 20 | 1 | | | |
| Ref. [22] | 3638 | 7.98 | 28.08 | 0.798 | 10.098 | 96.132 | 125.64 |
| Ref. [21] | 1852 | 7.65 | 27.76 | 0.835 | 10.247 | 45.581 | 126.46 |
| Ref. [23] | 555 | 7.55 | 27.91 | 0.741 | 10.476 | 26.862 | 110.53 |
| Ref. [24] | 1074 | 7.89 | 28.31 | 0.841 | 9.831 | 72.192 | 126.24 |
| Ref. [25] | 4932 | 7.81 | 28.13 | 0.785 | 10.531 | 73.268 | 130.08 |
| Ref. [7] | 3275 | 7.95 | 27.88 | 0.759 | 10.151 | 93.12 | 125.92 |
| Ref. [26] | 6753 | 7.54 | 28.03 | 0.757 | 9.371 | 47.858 | 105.01 |
| Ref. [27] | 3316 | 7.84 | 28.45 | 0.886 | 10.211 | 97.249 | 126.65 |
| **Ours** | **3045** | **8.25** | **28.75** | **0.931** | **10.515** | **81.674** | **125.35** |

**Table 5. The evaluation index data of Tian 'anmen Square images processed by various denoising algorithms.**

| Method | AL | Entropy | PSNR | SSIM | RMSE | AME | SDEM |
|---|---|---|---|---|---|---|---|
| Original | 254 | 7.26 | 20 | 1 | | | |
| Ref. [22] | 251 | 6.78 | 27.34 | 0.665 | 10.23 | 57.74 | 105.04 |
| Ref. [21] | 531 | 7.42 | 28.14 | 0.738 | 10.12 | 122.33 | 95.67 |
| Ref. [23] | 360 | 7.48 | 29.51 | 0.856 | 8.27 | 148.97 | 105.54 |
| Ref. [24] | 1914 | 7.58 | 28.11 | 0.734 | 10.26 | 71.46 | 121.04 |
| Ref. [25] | 452 | 7.19 | 28.53 | 0.941 | 9.39 | 101.93 | 97.48 |
| Ref. [7] | 442 | 7.74 | 27.69 | 0.817 | 10.63 | 60.83 | 97.59 |
| Ref. [26] | 189 | 7.20 | 31.66 | 0.904 | 10.47 | 134.74 | 131.95 |
| Ref. [27] | 596 | 6.91 | 27.87 | 0.827 | 10.13 | 209.92 | 94.24 |
| **Ours** | **839** | **7.59** | **28.96** | **0.635** | **10.09** | **68.10** | **126.14** |

According to the methods described in references [21, 23, 24, 25, 27], these methods are all based on multi-scale Retinex theory, with differences in preprocessing and processing methods for reflection and illumination components.

The method proposed in the literature [21] adopts the Brightness enhancement function (BEF) and improved Adaptive Contrast enhancement function (IACE), which significantly improves the overall brightness of the image. The texture rendering effect is excellent, especially with a close range, and the structural similarity with the original image is high. However, this method has the problem of stability, which may lead to over-whiteness of the image and loss of details when dealing with the perspective, and noise may be introduced in the fusion process, resulting in discontinuous boundaries.

The method proposed in reference [23] uses the bilateral filter to smooth the noise in the image effectively, and the visual effect is good. It also avoids the problem of discontinuous boundaries and preserves the structure of the image. However, the information entropy index of this method could be low, and less detailed information is retained.

The method proposed in reference [24] combines histogram equalization, bilateral gamma function correction, and bilateral filter to process reflection components. The processed image has a bright color, effectively removes noise, and reduces the appearance of halo and artifacts. Compared with the above methods [21, 23], the proposed method improves the information entropy and retains more detailed information. However, the processing effect of low-light images is unstable, there is a risk of excessive color distortion, and the difference from the original image is significant.

The method proposed in reference [25] uses the smoothing operator of the weighted least squares method for weighted fusion processing. This method reduces the phenomenon of brightness over-saturation and better preserves the image's structure. The visual effects in the waterfall and bench images perform well. However, the noise reduction effect is not apparent in the bench image. The effect of this method is not stable in images of different scenes and needs to be further adjusted.

The method proposed in reference [27] uses gamma correction and Wender-Fechner law to deal with the illumination components' low and high light areas, respectively. This method significantly improves the brightness and contrast of each region in the non-uniform illumination image and retains more details in the solid light area. However, this method will lose the structure of the image.

According to the research in the reference [7], automatic color equalization processing by applying an edge suppression mechanism and Gaussian distribution function can significantly

improve the brightness and contrast of the image, thus providing a better overall visual effect. However, this method may suffer from oversaturation in the close-range region, resulting in color bias towards darker tones. In addition, the proposed method could perform better in structural similarity index, peak mean square error, and square absolute error value, and the structure difference between the proposed method and the original image is large, significantly impacting the image quality.

According to reference [26], the Particle Swarm Optimization (PSO) algorithm performs well in gamma correction, which can effectively smooth the noise, improve the image's overall brightness, and better maintain the structure of the image. However, this method also has some shortcomings, such as the overall whiteness of the image and the loss of some detail information.

The processing method proposed in this paper performs well in visual and drying effects, and the image is bright in color, effectively reducing the excessive enhancement and gradient effect in the processing process. At the same time, the detailed information of the image is well preserved, and the critical texture information is also more prominent. After rigorous experimental comparison, the proposed method performs well in peak signal-to-noise ratio and ranks among the top three. However, in terms of indicators such as peak mean square error and absolute square error, the values are slightly lower in the middle. This shows that the proposed method can effectively reduce the difference with the original image and improve the quality of the reconstructed image when processing the image. In terms of structural similarity, the proposed method also achieved remarkable results. The mean square error of structural similarity is in the lower middle level, and the structural similarity index is among the best. This indicates that the processed image is more similar in structure to the original image. However, the structural similarity index has a low value when dealing with the Tiananmen smog image. This may be because the low-light yellowish haze image has more complex color and illumination changes, making it challenging to improve the evaluation index of the structural similarity index at the same time when enhancing denoising. However, this does not affect the excellent performance of the proposed method in most cases, which can ensure the quality and structure of the image.

## 3.4 Running time

The methods introduced in this paper include the automatic color equalization algorithm (ACE), Otsu's thresholding method, and the gradient P-M model. In the automatic color equalization algorithm(ACE), color space conversion, equalization filter, and inverse color space conversion constitute the whole processing flow. In the phase of color space conversion, the time complexity of the calculation is $O(MN)$. The filtering operation of the equalization filter involves the weighted sum of pixels and the comparison of pixel values, and its time complexity is related to the size of the filter($K*K$) and the number of pixels in the image. The time complexity of the filter can be approximated as $K^2O(MN)$. Therefore, the total time complexity of the ACE algorithm is shown in Eq (19):

$$ACE = 2O(MN) + K^2O(MN) \approx K^2O(MN) \qquad (19)$$

The Otsu's thresholding algorithm is based on histogram statistics, and its time complexity is $O(MN)$. In order to find the optimal threshold to maximize the between-class variance, the algorithm needs to iterate over all the thresholds. Therefore, the total time complexity of

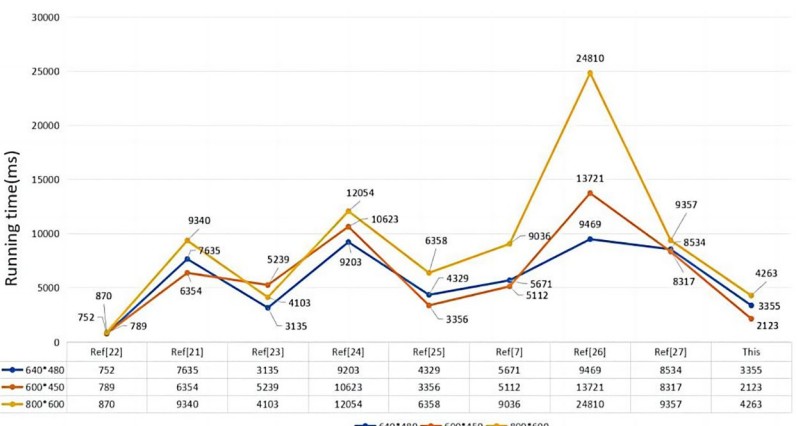

**Fig 7. Average running time line graph of different algorithms.**

Otsu's algorithm is shown in Eq (20):

$$\text{OTSU} = LO(MN) \tag{20}$$

The P-M model involves the calculation of the objective function, the calculation of the gradient, and the update of the parameters in each iteration. The computational complexity of the objective function and the gradient are $O(MN)$, while the updating complexity of the parameters is $O(1)$. The total time complexity of the gradient P-M model can be approximated as shown in Eq (21):

$$\text{PM} = TO(MN) + TO(MN) + TO(1) \approx TO(MN) \tag{21}$$

Therefore, the total time complexity of the proposed method is shown in Eq (22):

$$\text{Toatl complexity} = K^2 O(MN) + LO(MN) + TO(MN) \tag{22}$$

Where $K$ is the size of the equalization filter, $M$ represents the image's height, $N$ represents the image's width, and $L$ represents the number of gray levels of the image.

An algorithm running time evaluation metrics are significant for user experience, resource utilization and cost-effectiveness, scalability and adaptability, algorithm comparison and selection, etc. Fig 7 shows the average running time (in milliseconds) of 20 executions of different methods on images of three different resolutions in the experimental environment of this paper. According to the data in Fig 7, compared with the denoising algorithms in [7, 21, 24–27], the proposed method can process noisy images faster and output denoised image data. Compared with [22, 23], the processing speed is lower, but the quality of the output image is better than them.

## 3.5 Ablation experiment

In order to verify the influence of each part of the model on the experimental results, ablation experiments were performed on each part of the model method and the corresponding combination. The methods include 1) a single automatic color equalization algorithm; 2) a single P-M model; 3) Automatic color equalization +P-M model without gradient enhancement using the OTSU method; 4) The initial noise image is directly processed by OTSU's gradient P-M model; 5) The method proposed in this paper. The ablation experimental results of methods 1-5 in waterfall and bench images are shown in Tables 6 and 7, respectively.

According to the data in Tables 6 and 7, the average brightness of the image processed by method 1 shows excellent performance. However, regarding information entropy value, the images processed by method 2 show apparent advantages.

Method 3 combines Method 1 and Method 2. Compared with the two methods alone, the average brightness of method 3 is increased by 60% and 134%, respectively, and the information entropy value is increased by 0.7% and 4.4%, respectively. However, there are some problems in the noise distortion degree and pixel difference effect of the image, which may introduce gradient noise in the processing process. Nevertheless, the image maintains an excellent structure to ensure the accurate transmission of basic visual information.

Based on Method 2, Method 4 further introduces Otsu's algorithm and enhancement function. Compared with method 2, the average brightness of method 4 on the two images is increased by 37% and 102%, respectively, and the information entropy index is also increased by 1% and 5%, respectively. According to the data results of the last five evaluation indicators, the proposed method can more accurately present the details of the image structure and effectively improve the image quality. However, this method also has problems: the brightness structure will be lost in the processing.

Method 5, which further introduces the Otsu algorithm and the enhancement function based on Method 3, is proposed in this paper. Compared with method 3, the average brightness of method 5 on the two images is increased by 7% and 25%, and the information entropy value is increased by 1.8% and 4.9%, respectively. According to the peak signal-to-noise ratio, structural similarity index, structural similarity mean square error, and absolute square error data, the structural and perceptual quality of the image are significantly improved, and the brightness structure is more similar to the original image. However, it should be noted that part of the detailed structure may be lost during the processing. Considering the trade-off between different indicators, the improved scheme proposed in this paper can effectively improve the image contrast and retain the image details.

## 4. Conclusion

This study proposes a novel denoising method combining the Automatic Color equalization (ACE) algorithm and gradient P-M model (referred to as ACE-GPM for short), specifically for

**Table 6. Ablation experimental results for waterfall images.**

| Method | AL | Entropy | PSNR | SSIM | RMSE | AME | SDEM |
|---|---|---|---|---|---|---|---|
| ACE | 236 | 7.44 | 27.82 | 0.78 | 10.49 | 120.23 | 126.45 |
| P-M | 198 | 7.51 | 27.67 | 0.816 | 3.49 | 182.66 | 120.44 |
| ACE+P-M | 349 | 7.53 | 26.13 | 0.839 | 10.54 | 57.63 | 124.47 |
| OTSU+P-M | 273 | 7.59 | 27.36 | 0.793 | 10.47 | 105.32 | 116.82 |
| **This paper** | **376** | **7.77** | **27.78** | **0.841** | **10.39** | **87.11** | **123.43** |

**Table 7. Ablation experimental results for bench images.**

| Method | AL | Entropy | PSNR | SSIM | RMSE | AME | SDEM |
|---|---|---|---|---|---|---|---|
| ACE | 1261 | 7.53 | 27.33 | 0.823 | 10.06 | 86.66 | 128.77 |
| P-M | 824 | 7.62 | 27.74 | 0.795 | 2.76 | 79.39 | 107.51 |
| ACE+P-M | 2429 | 7.86 | 28.32 | 0.882 | 10.73 | 82.63 | 127.56 |
| OTSU+P-M | 1669 | 8.03 | 28.03 | 0.839 | 10.19 | 77.16 | 127.23 |
| **This paper** | **3045** | **8.25** | **28.75** | **0.931** | **10.51** | **81.67** | **125.35** |

low-illumination hazy noisy images. This method first uses automatic color balance to enhance image contrast. Then, it applies OTSU threshold segmentation based on the most between-cluster variance method and threshold, leading to the image's gradient field enhancement. Finally, the gradient of P—M removes the noise-denoising model, effectively reserving the detail of the image and texture information simultaneously. Experimental results show that the proposed method retains more detailed texture information while improving image contrast and removing noise. It also has significant advantages in running time and image quality evaluation index.

In addition, the critical parameters of the P-M model are accurately determined by manual iteration to ensure the convergence and stability of the algorithm. Although selecting these parameters is crucial to the efficiency of the gradient descent method, there is still room for further optimization, especially the adaptive parameter adjustment mechanism under different pixel densities and image sharpness conditions. Future research will explore the adaptive learning rate strategy in the deep learning environment to realize the adaptive optimization of the gradient P-M model parameters so as to enhance the generalization ability and practicability of the model and further improve its performance on various evaluation indicators.

## Author Contributions

**Conceptualization:** Wuyi Li, Guanglu Zhou, Xingjian Wang.

**Data curation:** Wuyi Li.

**Formal analysis:** Wuyi Li.

**Funding acquisition:** Xingjian Wang.

**Investigation:** Wuyi Li, Xingjian Wang.

**Methodology:** Wuyi Li.

**Software:** Wuyi Li.

**Supervision:** Xingjian Wang.

**Writing – original draft:** Wuyi Li, Guanglu Zhou.

**Writing – review & editing:** Wuyi Li, Guanglu Zhou, Xingjian Wang.

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
