## [Decision Letter · Decision Letter 0]

19 Oct 2023

PONE-D-23-30473Improved Image Enhancement by Partial Differential Equation and Automatic Color EqualizationPLOS ONE

Dear Dr. wang,

Thank you for submitting your manuscript to PLOS ONE. After careful consideration, we feel that it has merit but does not fully meet PLOS ONE’s publication criteria as it currently stands. Therefore, we invite you to submit a revised version of the manuscript that addresses the points raised during the review process.

We look forward to receiving your revised manuscript.

Kind regards,

Sen Xiang

Academic Editor

PLOS ONE

Journal Requirements:

4. Thank you for stating the following financial disclosure: "This work was supported in part by Natural Science Foundation of Heilongjiang Province under Grant LH2020C048, and in part by the Harbin Science and Technology Innovation Talent Research Foundation  under Grant 2017RAQXJ108."

5. Thank you for stating the following in the Acknowledgments Section of your manuscript: "This work was supported in part by Natural Science Foundation of Heilongjiang Province under Grant LH2020C048, and in part by the Harbin Science and Technology Innovation Talent Research Foundation under Grant 2017RAQXJ108."

Please remove any funding-related text from the manuscript and let us know how you would like to update your Funding Statement. Currently, your Funding Statement reads as follows: "This work was supported in part by Natural Science Foundation of Heilongjiang Province under Grant LH2020C048, and in part by the Harbin Science and Technology Innovation Talent Research Foundation  under Grant 2017RAQXJ108."

Reviewers' comments:

Reviewer's Responses to Questions

**Comments to the Author**

1. Is the manuscript technically sound, and do the data support the conclusions?

Reviewer #1: Yes

Reviewer #2: Yes

Reviewer #3: Yes

2. Has the statistical analysis been performed appropriately and rigorously? 

Reviewer #1: No

Reviewer #2: Yes

Reviewer #3: Yes

3. Have the authors made all data underlying the findings in their manuscript fully available?

Reviewer #1: Yes

Reviewer #2: Yes

Reviewer #3: Yes

4. Is the manuscript presented in an intelligible fashion and written in standard English?

Reviewer #1: No

Reviewer #2: Yes

Reviewer #3: Yes

5. Review Comments to the Author

Reviewer #1: Abstract:

1. Too much general abstract, it needs to be improved; how the methodology increased the quality of the reconstructed image and which state of the art methods are used.

2. The main methodology should come direct after the general sentences of the abstract.

3. The title looks too general, it seems to be a sub-section in academic paper.

3. Add partial of your results to the last part of the Abstract

Introduction: The introduction should discuss more about the contaminated images and types of noise that attack the digital images such as additive, shot and multiplicative noise. Studies such as the following references should be included in the introduction or related work sections:

Natural digital image mixed noise removal using regularization Perona–Malik model and pulse coupled neural networks. Soft Comput (2023).

Additive Gaussian noise removal based on generative adversarial network model and semi-soft thresholding approach. Multimedia tools and application. Springer 2022.

Natural image deblurring using recursive deep convolutional neural network (R-DbCNN) and Second-Generation Wavelets. 2019 IEEE International Conference on signal and image processing applications (ICSIPA).

Methodology: The contributions of the paper are not clear. It needs to be focused on the maincontribution that the study brought it out. The use of (PM diffusion equation) is not new method. It has its demerits in smoothing the enhanced image.

The use of discrete Fourier transform has its negative imapct in its complicated structure of the method, discus its impact and show how it can work for your method.

It is very difficult to follow either the topic is image denoising or image enhancement. Make it clear in the manuscript.

Please draw block diagram of the proposed work and more detailed explanations are needed.

The experiments:

1.Firstly, should conduct more experiments, such as, a group of visual comparisonsor intermediate results, to verify the effectiveness of the proposed method.

2. Experimental setup: From this paper the repeatability of the experiments is very limited given the missing details. It is not clear if the authors used a reimplementation of the approaches or if they use code provided by related work.Also Hardwar specifications and software platforms must be mentioned in the manuscript.

3. It is better to focus on one kind of noise effect rather more than one. May the proposed method has significant performance on Gaussian noise more than others.

4. In experiments, the authors should provide more detailed information about the construction of training and testing sets. You need to add visual benchmark images to evaluate the proposed method subjectively.

5. Add FOM figure of merit as a quantitative assessment to add more reliability performance to the proposed method.

6. What is the effect of the running time in general and at several range of noise levels? May drawing curve will show the effectiveness of the performance more than what the table did.

Hardware and software specifications of your system must be added to the text in order to fair judge the proposed method.

7. The computational complexity of the method needs to be evaluated.

8. Otsu threshold value needs to be more elaborated.

Presentation of the manuscript: The way of presentation of this work is need to be improved, the graphs and tables really need to be focused and zoomed.

Reference and Conclusion : References need to be updated. The conclusion section should make clear the future work part and criticize the proposed method where the limitation is, and how it can be solved in future studies.

Reviewer #2: Authors used partial differential equation and automatic color equalization for contrast enhancement. The paper is technically well represented but fails in the following points.

It will be good for the reader if the previous works are represented in the form of a table with their pros and cons. Some of the recent works in the same domain have not cited and mentioned below.

D. Vijayalakshmi, and Malaya Kumar Nath, "A strategic approach towards contrast enhancement by two-dimensional histogram equalization based on total variational decomposition", Multimedia Tools and Applications: Springer Nature (IF - 2.396, indexed in SCIE), pp: **, vol. **, no. **, October 2022. https://doi.org/10.1007/s11042-022-13932-7

D. Vijayalakshmi, and Malaya Kumar Nath, "A novel multilevel framework based contrast enhancement for uniform and non-uniform background images using Suitable Histogram Equalization", Digital Signal Processing: Elsevier (IF - 2.92, indexed in SCIE), vo1 127, pp. 103532 (1-19), July 2022. https://doi.org/10.1016/j.dsp.2022.103532

D. Vijayalakshmi, and Malaya Kumar Nath, "A novel contrast enhancement technique using gradient based joint histogram equalization", Circuits, Systems and Signal Processing: Springer Nature (IF - 2.225, indexed in SCIE), pp: 3929-3967, vol. 40, no. **, February 2021. https://doi.org/10.1007/s00034-021-01655-3

D. Vijayalakshmi, and Malaya Kumar Nath, “A compendious analysis of advances in HE methods for contrast enhancement”, 2nd International Conference on VLSI, Communication and Signal Processing (VCAS) 2019, 21-23 October 2019, vol. 683, pp. 325-346 (Published in Advances in VLSI, Communication and Signal Processing, Lecture Notes in Electrical Engineering, Springer, Singapore, Ch. No: 26), NIT Allahabad, Prayagraj, India. https://doi.org/10.1007/978-981-15-6840-4_26

The performance measures need to be evaluated without any reference (or ground truth).

In many locations the typo and grammatical errors are noticed and need to be thoroughly checked. Rc(p), Ic(p), Ic(j) are not represented in the same way in equation 1 and in the text.

It will be good for representing the steps for the proposed approach for better understanding by the reader.

The references in the reference section are not uniform (i.e, name of author, page no, title, vol, journal name, DOI etc).

More figures need to be presented for better understanding of the approach.

It will be good if different clarity (low contrast, high contrast, bright) images are applied to the algorithm for analyzing its performance.

Reviewer #3: This paper improves the standard PM model based on Partial Differential equations (PDEs). Firstly, based on the PM diffusion equation, a convergence enhancement function is added to improve the image gradient. Then, Otsu's method was used for threshold segmentation to determine the diffusion threshold parameters of the model. Finally, discrete Fourier transform was used instead of convolution to improve the execution efficiency of ACE. A large number of experimental results show that the improved algorithm and model improve the execution efficiency, reduce the noise, enhance the image boundary and the overall visual effect, and eliminate the atomization. The performance of this method has been proved by experiments, but there are still some problems in the paper:

1. For "summary", it should be straightforward. The authors need to re-refine the abstract by first pointing out the challenging problems, then specific solutions, and finally experimental results. Ensure that the abstract concisely summarizes the paper in the abstract and citation services. Keep the abstract between 150 and 180 words long. (see this article: Underwater image enhancement by attenuated color channel correction and detail preserved contrast enhancement ")

2. In the introduction section, the paper does not clearly highlight the unique contribution of this paper. The author should summarize the contribution in three points to summarize the content of the paper simply and accurately.

3. The references in the introduction section of this paper are not novel enough. Authors are advised to cite more articles published within the last three years to maintain the timeliness and relevance of the literature review. (Please refer to these articles: Underwater image enhancement via piecewise color correction and dual prior optimized contrast enhancement, An Image Restoration Method With Generalized Image Formation Model for Poor Visible Conditions)

4. There are many format problems in this paper, some have colon at the end of the formula and some do not, and the combination of multiple letters in the formula, italics represents the multiplication of multiple variables, otherwise it should be in the form of text, please check and revise the full text. The variables in the formula are not explained in many places by the author. Please add them. (Please refer to this article: "Underwater Image Enhancement via Weighted Wavelet Visual Perception Fusion")

5. This paper lacks a paper framework flowchart, and in order to improve the understandability of the paper, it is recommended that the author add a complete flowchart to clearly present the methodological framework and steps of the research.

6. The images of Figure 1, Figure 2, Figure 3, Figure 4_1 and Figure 4_2 in this paper are blurred, and it is suggested that the author provide high-resolution images to improve the clarity of the images.

7. This paper lacks ablation experiments in the experimental part, and the author needs to add ablation experiments that reflect the advantages of the improved PM model.

6. PLOS authors have the option to publish the peer review history of their article (what does this mean?). If published, this will include your full peer review and any attached files.

Reviewer #1: No

Reviewer #2: No

Reviewer #3: No

---

## [Author Response · Author response to Decision Letter 0]

6 Dec 2023

Reviewer 1: 

I. Abstract: 

1. The updated abstract reflects this paper’s image reconstruction method steps. Firstly, an automatic color equalization algorithm(ACE) is used to improve the contrast of the low-light image containing noise. Secondly, the maximum betweencluster variance method(OTSU) was used for threshold segmentation. Then, according to the segmentation threshold, the image was enhanced by gradient using the enhancement function. Finally, the P-M model was used to denoise and enhance the gradient image. 

2. The method proposed in this paper is mainly elaborated at the beginning of the third sentence of the abstract. 

3. The title is updated to:Low illumination fog noise image denoising method based on ACE-GPM. 

4. The comparison results between the proposed method and other methods are reflected in the abstract. Compared with other denoising methods, the ACE-GPM method proposed in this paper improves image contrast and removes noise while effectively retaining image details and texture information. The information entropy value is increased by 0.42 on average. In addition, the proposed method requires fewer computational resources while maintaining image quality. 

II. Introduction: 

In the introduction of the revised version, this paper first expounds on the common types of noise in digital images and the reasons for their generation. Subsequently, we explain the denoising methods proposed in the literature and their advantages and disadvantages. Finally, we show the proposed method’s working block diagram and method steps so readers can better understand our method.

III. Method: 

1. In the ninth paragraph of the introduction, this paper elaborates on the anisotropic P-M diffusion equation model and its advantages and disadvantages in image denoising and enhancement. This model has a wide range of application values in image processing, but due to its limitations, some areas also need improvement. 

2. In the research method section of this paper, part D elaborates in-depth, and how to apply the OTSU method for gradient enhancement is introduced in detail. At the same time, we also clarify how the method ADAPTS to the P-M model and its embodiment of the performance improvement of the P-M model. 

3. At the end of the introduction, this paper shows a simple working block diagram of the proposed method and elaborates each step in detail. 

IV. Experiment: 

1. In the revised manuscript’s experimental part, this paper adds two sets of visual contrast and FOM merit evaluation to verify the effectiveness and reliability of the proposed method comprehensively. These additional experimental groups are intended to provide more exhaustive empirical support for our approach. 

2. The revised manuscript details the hardware and software specifications in Section A of the experiment. The introduction section clearly describes the working steps of the proposed method. Each part of the research method constitutes the processing flow of the method in turn. 

3. This paper studies the noise reduction problem of low-light fog images. We propose an effective denoising method for fog noise in low-light fog images. 

4. This paper selects low-light fog noise images taken under natural conditions, and the comparison results between visual benchmark images and images processed by the method are provided in Section D of the experiment. 

5. In Table 2 of the original paper, we detail the objective index evaluation of the images processed by various methods, including indicators such as average brightness, standard deviation, information entropy, peak signal-to-noise ratio (PSNR), and similarity index (SSIM). Because these evaluation metrics are not explained in detail in the original paper, it may have brought you questions. Therefore, in Section B of the experimental section of the revised manuscript, we present the FOM merit evaluation index and its calculation formula in this paper. In Tables 3, 4, and 5 in Section D, we detail the FOM merit evaluation index data of the processed images by each denoising algorithm. 

6. According to the data in Table 2 of the original paper, the running time of vari- ous methods for processing waterfall images may need to be more intuitive to show their effects. Therefore, after expert advice, this paper shows the line chart of running time for images with different resolutions in FIG. 7, part E of the experiment of the revised manuscript. This gives the reader a clearer picture of how different processing methods differ in running time and helps the reader evaluate the performance of each method more accurately. 

7. The revised version of experiment section E shows the evaluation data of each part of the proposed method and the total time complexity. 

8. In the previous version, the introduction of OTSU’s algorithm may have some things that could be improved. Based on the expert feedback, this paper gives a more comprehensive description of the OTSU threshold method. It explains in detail how it is applied to the P-M model in Section B of the revised version of research methods. 

9. With revisions and updates, the references and conclusions in this paper have been refined. In the references, we included some crucial articles published recently to ensure the timeliness and relevance of this paper. At the same time, in the conclusion section, we point out some limitations of the proposed method. We still need to adaptively select the P-M model’s critical parameters for different pixel and definition images.

Reviewer 2:

A: In the revised version of Experiment Section D, we show the contrast effect clearly in the form of a table. See Table 2 for details.

B: In the experiment D section of the revised draft, we conduct multiple comparison experiments of eight existing denoising algorithms. We use the visual effect and FOM figure of merit as evaluation indicators to compare and evaluate the performance of various methods.

C: The format of the references has been corrected, and references have been made to recent journal articles to ensure the timeliness and relevance of this article. 

D: For readers’ convenience, this paper’s work block diagram is provided in the revised draft, and the specific operation steps of each part are described in detail. Such an arrangement provides a more transparent and intuitive understanding, which helps readers better grasp the main content and ideas of this paper. 

E: The proposed method is currently suitable for low-light fog images, so images with different sharpness (low contrast, high contrast, brightness) have yet to be applied to the algorithm for performance evaluation. 

Reviewer 3: 

1. Abstract: We have revised and updated the paper’s abstract according to the expert advice. Firstly, this paper leads to the proposed method for the problems of the existing P-M model. Secondly, the implementation of the method proposed in this paper is briefly described. Finally, the experimental results comparing the proposed method with other denoising methods in the literature are described to prove the effectiveness of the proposed method. 

2. Introduction: In the revised manuscript’s introduction section, we describe this paper’s workflow in detail and clearly explain the tasks of each stage. In this way, we can more accurately communicate our work’s content, goals, and importance so that readers can better understand our work. 

3. References Cited: After revision, all references cited in this paper have been updated to ensure timeliness and relevance. 

4. Format and formula: After carefully reviewing and revising the first draft and referring to the reference articles provided by experts, we have revised the format and formula to ensure that the final revised draft meets the rigorous, stable, rational, and official requirements in form and content.

5. Flowchart of the paper framework: At the end of the introduction section of the revised manuscript, we attach a flowchart of the work of this paper, which details the tasks and objectives of each stage. 

6. Image clarity: In the revised version, to more fully demonstrate the effectiveness of the proposed method, we conduct more comparative experiments and improve the image’s contrast. 

7. Ablation Experiment: In the revised version, the ablation experiment is added to Part F of an experiment to evaluate the influence of each part of the method and the combination of each part on the P-M model to reflect the effectiveness of the improved strategy in this paper.

---

## [Decision Letter · Decision Letter 1]

18 Dec 2023

PONE-D-23-30473R1Low illumination fog noise image denoising method based on ACE-GPMPLOS ONE

Dear Dr. wang,

Thank you for submitting your manuscript to PLOS ONE. After careful consideration, we feel that it has merit but does not fully meet PLOS ONE’s publication criteria as it currently stands. Therefore, we invite you to submit a revised version of the manuscript that addresses the points raised during the review process.

We look forward to receiving your revised manuscript.

Kind regards,

Sen Xiang

Academic Editor

PLOS ONE

Reviewers' comments:

Reviewer's Responses to Questions

**Comments to the Author**

1. If the authors have adequately addressed your comments raised in a previous round of review and you feel that this manuscript is now acceptable for publication, you may indicate that here to bypass the “Comments to the Author” section, enter your conflict of interest statement in the “Confidential to Editor” section, and submit your "Accept" recommendation.

Reviewer #1: (No Response)

Reviewer #2: All comments have been addressed

Reviewer #3: (No Response)

2. Is the manuscript technically sound, and do the data support the conclusions?

Reviewer #1: Partly

Reviewer #2: Partly

Reviewer #3: Yes

3. Has the statistical analysis been performed appropriately and rigorously? 

Reviewer #1: No

Reviewer #2: Yes

Reviewer #3: Yes

4. Have the authors made all data underlying the findings in their manuscript fully available?

Reviewer #1: Yes

Reviewer #2: Yes

Reviewer #3: Yes

5. Is the manuscript presented in an intelligible fashion and written in standard English?

Reviewer #1: Yes

Reviewer #2: Yes

Reviewer #3: Yes

6. Review Comments to the Author

Reviewer #1: . Comments to the Author

I have the following problems with this paper:

- The Introduction can be revised to emphasize the main contribution of the work.

- The architecture of the framework needs to be described in more detail. I would suggest the authors present more graphical information.

- For the experiments, some discussion should be given about the results to explain the merits and drawbacks of the proposed method.

- The authors did not a present full objective quality comparison on test images to check the quality of their methodology. Maybe authors will present an estimation of RMSE, PSNR, AME, EMEE, SDME, Visibility, TDME, BIQI, BRISQUE, ILNIQE, or NIQE.

- To verify the effective of the proposed enhancement method, it is better to compare several representative state-of-the-art image methods.

The introduction should discuss more deep in the chronological techniques and the up to date noise removal methods in different approaches. Studies such as the following references should be included in the introduction or related work sections:

Natural image noise removal using nonlocal means and hidden Markov models in transform domain. The Visual Computer. Springer. 34. 1661-1675. 2018

Additive Gaussian noise removal based on generative adversarial network model and semi-soft thresholding approach. Multimedia tools and application. Springer 2022.

Natural digital image mixed noise removal using regularization Perona–Malik model and pulse coupled neural networks. Soft Computing. 2023

Reviewer #2: The manuscript uses the ACE-GPM methods for fog noise image denoising. Authors have well addressed the queries raised by the reviewers. But, still some care is needed to make the manuscript free from any typo and grammatical errors. In many cases the abbreviations are not defined at proper position. Some cases the space was missing. The paragraphs are not justified. The block diagram needs more explanation. How this model performs in presence of noise was not discovered? The approach needs to be compared with the recently developed methods and mentioned below.

https://doi.org/10.1016/j.gmod.2023.101206

https://doi.org/10.1007/s11042-022-13932-7

https://doi.org/10.1016/j.dsp.2022.103532

https://doi.org/10.1007/s00034-021-01655-3

Can you share the code and data for the reproducibility of the work?

Reviewer #3: This paper proposes an image denoising algorithm (ACE-GPM) based on the Automatic Color Equalization algorithm (ACE) and the Gradient P-M model. Firstly, ACE is utilized to enhance the contrast of low-light, foggy, and noisy images. Secondly, OTSU is applied for threshold segmentation, accurately identifying distinct regions. Subsequently, an enhancement function is employed to boost foreground and background gradients differently, emphasizing edges and details. Finally, the Gradient P-M model is used, employing gradient descent to denoise and enhance the image, highlighting edges and details more effectively. Experimental results demonstrate the effectiveness of the proposed method in image denoising. However, certain issues are identified through the experiments：

1. The methods in related fields introduced in this article in "Introduction" are not novel enough, please add some latest methods. (Specific reference and quote: “Weighted Wavelet Visual Perception Fusion”、“Minimal Color Loss and Locally Adaptive Contrast Enhancement”and“Piecewise Color Correction and Dual Prior Optimized Contrast Enhancement”).

2. The flowchart needs to be redrawn and some visual aids should be added accordingly. The total flowchart needs to include enhanced images of the results, and a step-by-step description of the entire algorithmic workflow should be included at the bottom of the flowchart, and the authors should be asked to optimize it.

3. The parameters in equation (6) of the article need to be given a hyperparametric analysis. (Reference can be made to: " Minimal Color Loss and Locally Adaptive Contrast Enhancement”).

4. As for the experimental part, there is a lack of comparative experiments with the latest enhancement methods.

5. In the formula of the article, when multiple letters of a variable are together, they are in block type instead of italic type, and the expression of the formula is incorrect, so the authors are requested to double-check and revise it.

6. The drawings in the paper are not clear, so high-resolution drawings need to be provided.

7. PLOS authors have the option to publish the peer review history of their article (what does this mean?). If published, this will include your full peer review and any attached files.

Reviewer #1: No

Reviewer #2: **Yes: **MALAYA KUMAR NATH

Reviewer #3: No

---

## [Author Response · Author response to Decision Letter 1]

1 Jan 2024

Thank you for taking time out of your busy schedule to read my manuscript and for your reply letter. For the questions and opinions of the experts, I will reply one by one.

Given the problem that Expert 1 and Expert 3 pointed out that they did not receive relevant responses in the last round of review, we at this moment make the following remarks: In the previous submission, the PDF called "Response to Reviewers" had pages 1 and 2 in response to Expert 1's review, and page 4 in response to expert 3's review. Since the experts could not see the corresponding reply, I will attach the related reply content from the last time for your reference in this reply. First, the response to the previous round of expert 1's opinion is as follows:

First, the response to the previous round of expert 1's opinion is as follows:

一、Abstract： 

1. The updated abstract reflects this paper's image reconstruction method steps. Firstly, an 

automatic color equalization algorithm(ACE) is used to improve the contrast of the low-light 

image containing noise. Secondly, the maximum between-cluster variance method(OTSU) was 

used for threshold segmentation. Then, according to the segmentation threshold, the image was 

enhanced by gradient using the enhancement function. Finally, the P-M model was used to denoise 

and enhance the gradient image. 

2. The method proposed in this paper is mainly elaborated at the beginning of the third sentence 

of the abstract. 

3. The title is updated to ： Low illumination fog noise image denoising method based on 

ACE-GPM 

4. The comparison results between the proposed method and other methods are reflected in the 

abstract. Compared with other denoising methods, the ACE-GPM method proposed in this paper 

improves image contrast and removes noise while effectively retaining image details and texture 

information. The information entropy value is increased by 0.42 on average. In addition, the 

proposed method requires fewer computational resources while maintaining image quality. 

二、Introduction： 

In the introduction of the revised version, this paper first expounds on the common types of noise 

in digital images and the reasons for their generation. Subsequently, we explain the denoising 

methods proposed in the literature and their advantages and disadvantages. Finally, we show the 

proposed method's working block diagram and method steps so readers can better understand our 

method. 

三、Method： 

1. In the ninth paragraph of the introduction, this paper elaborates on the anisotropic P-M 

diffusion equation model and its advantages and disadvantages in image denoising and 

enhancement. This model has a wide range of application values in image processing, but due to 

its limitations, some areas also need improvement. 

2. In the research method section of this paper, part D elaborates in-depth, and how to apply the 

OTSU method for gradient enhancement is introduced in detail. At the same time, we also clarify 

how the method ADAPTS to the P-M model and its embodiment of the performance improvement 

of the P-M model. 

3. At the end of the introduction, this paper shows a simple working block diagram of the 

proposed method and elaborates each step in detail. 

四、Experiment： 

1. In the revised manuscript's experimental part, this paper adds two sets of visual contrast and 

FOM merit evaluation to verify the effectiveness and reliability of the proposed method 

comprehensively. These additional experimental groups are intended to provide more exhaustive 

empirical support for our approach. 

2. The revised manuscript details the hardware and software specifications in Section A of the 

experiment. The introduction section clearly describes the working steps of the proposed method.Each part of the research method constitutes the processing flow of the method in turn. 

3. This paper studies the noise reduction problem of low-light fog images. We propose an 

effective denoising method for fog noise in low-light fog images. 

4. This paper selects low-light fog noise images taken under natural conditions, and the 

comparison results between visual benchmark images and images processed by the method are 

provided in Section D of the experiment. 

5. In Table 2 of the original paper, we detail the objective index evaluation of the images 

processed by various methods, including indicators such as average brightness, standard deviation, 

information entropy, peak signal-to-noise ratio (PSNR), and similarity index (SSIM). Because 

these evaluation metrics are not explained in detail in the original paper, it may have brought you 

questions. Therefore, in Section B of the experimental section of the revised manuscript, we 

present the FOM merit evaluation index and its calculation formula in this paper. In Tables 3, 4, 

and 5 in Section D, we detail the FOM merit evaluation index data of the processed images by 

each denoising algorithm. 

6. According to the data in Table 2 of the original paper, the running time of various methods for 

processing waterfall images may need to be more intuitive to show their effects. Therefore, after 

expert advice, this paper shows the line chart of running time for images with different resolutions 

in FIG. 7, part E of the experiment of the revised manuscript. This gives the reader a clearer 

picture of how different processing methods differ in running time and helps the reader evaluate 

the performance of each method more accurately. 

7. The revised version of experiment section E shows the evaluation data of each part of the 

proposed method and the total time complexity. 

8. In the previous version, the introduction of OTSU's algorithm may have some things that could 

be improved. Based on the expert feedback, this paper gives a more comprehensive description of 

the OTSU threshold method. It explains in detail how it is applied to the P-M model in Section B 

of the revised version of research methods. 

9. With revisions and updates, the references and conclusions in this paper have been refined. In 

the references, we included some crucial articles published recently to ensure the timeliness and 

relevance of this paper. At the same time, in the conclusion section, we point out some limitations 

of the proposed method. We still need to adaptively select the P-M model's critical parameters for 

different pixel and definition images.

Secondly, the response to Expert 3's last round of comments is as follows:

1. Abstract：We have revised and updated the paper's abstract according to the expert advice. Firstly, this paper leads to the proposed method for the problems of the existing P-M model. Secondly, the implementation of the method proposed in this paper is briefly described. Finally, the experimental results comparing the proposed method with other denoising methods in the literature are described to prove the effectiveness of the proposed method.

2. Introduction：In the revised manuscript's introduction section, we describe this paper's workflow in detail and clearly explain the tasks of each stage. In this way, we can more accurately communicate our work's content, goals, and importance so that readers can better understand our work.

3. References Cited: After revision, all references cited in this paper have been updated to ensure timeliness and relevance.

4. Format and formula: After carefully reviewing and revising the first draft and referring to the reference articles provided by experts, we have revised the format and formula to ensure that the final revised draft meets the rigorous, stable, rational, and official requirements in form and content.

5. Flowchart of the paper framework: At the end of the introduction section of the revised manuscript, we attach a flowchart of the work of this paper, which details the tasks and objectives of each stage.

6. Image clarity: In the revised version, to more fully demonstrate the effectiveness of the proposed method, we conduct more comparative experiments and improve the image's contrast.

7. Ablation Experiment: In the revised version, the ablation experiment is added to Part F of an experiment to evaluate the influence of each part of the method and the combination of each part on the P-M model to reflect the effectiveness of the improved strategy in this paper.

Finally, a reply to the opinions of the experts in this round:

Review 1：

一、Comments on the Introduction: 

1. According to your professional suggestions, we have carefully revised the middle part of the third page of the revised manuscript to focus on the core contributions of this paper. In addition, we have redrawn the system block diagram and refer to the paper "Underwater Image Enhancement by Attenuated Color Channel Correction and Detail Preserved Contrast. The overview diagram of the method in Enhancement "draws a block diagram of the working of the proposed method. The work block diagram shows the graphical information of each step in detail to help readers understand the research content of this paper more intuitively. 

2. In the introduction section, we discuss five representative noise-denoising algorithms in-depth. The following is the literature corresponding to these denoising algorithms:

1. Paper 1：Natural digital image mixed noise removal using regularization Perona–Malik model and pulse coupled neural networks.Soft Computing. 2023: 1-10. 

2. Paper 2：Underwater image enhancement by attenuated color channel correction and detail preserved contrast enhancement. IEEE Journal of Oceanic Engineering 2022, 47(3): 718-735. 

3. Paper 3：Natural image noise removal using nonlocal means and hidden Markov models in transform domain. The Visual Computer. 2018, 34: 1661-1675. 

4. Paper 4：Underwater image enhancement via piecewise color correction and dual prior optimized contrast enhancement. IEEE Signal Processing Letters. 2023, 30: 229-233. 

5. Paper 5：Additive Gaussian noise removal based on generative adversarial network model and semi-soft thresholding approach. Multimedia Tools and Applications.2023, 82(5): 7757-7777. 

二、Comments on the experiment:

1. According to your advice from experts, to objectively evaluate the quality of test images, we have added evaluation indicators such as peak mean square Error (RMSE), Square Absolute Error (AME), and mean square error of Structural Similarity (SDME). These metrics help to measure image quality more comprehensively, and their specific mathematical formulas have been detailed in Experiment Section B. With these improvements, we can evaluate the performance of various methods in terms of image quality more accurately. 

2. To verify the effectiveness of the proposed enhancement method, this paper compares the latest representative image enhancement methods and briefly describes their algorithm process, advantages, and disadvantages. The following is a list of literature corresponding to enhancement methods:

（1）Low-Light Image Enhancement Method Based on Retinex Theory by Improving Illumination Map. Applied Sciences. 2022, 12(10): 5257.

（2）Low-illumination image enhancement algorithm based on improved multi-scale Retinex and ABC algorithm optimization.Frontiers in Bioengineering and Biotechnology. 2022, 10: 865820.

（3）An adaptive low-illumination image enhancement algorithm based on weighted least squares optimization. Journal of Physics: Conference Series. IOP Publishing. 2022, 2181(1): 012011.

（4）Non-Uniform-Illumination Image Enhancement Algorithm Based on Retinex Theory. Applied Sciences. 2023, 13(17): 9535.

Review 2：

Based on your professional advice, I have reviewed the manuscript again and corrected some spelling and grammar errors. I have fixed abbreviations that need to be defined in appropriate places in the text. Given the system block diagram in the introduction section, I have referred to the paper Underwater Image Enhancement by Attenuated Color Channel Correction and Detail Preserved Contrast Enhancement, "which redraws the block diagram of the work and elaborates the working steps and the main contributions of this paper in the following. In addition, I have updated the comparative experiment method in the revised manuscript to address the problem that the comparative experiment method is not novel enough. I chose methods from recently published papers for comparative experiments, and the specific forms are as follows:

（1）Low-Light Image Enhancement Method Based on Retinex Theory by Improving Illumination Map. Applied Sciences. 2022, 12(10): 5257.

（2）Low-illumination image enhancement algorithm based on improved multi-scale Retinex and ABC algorithm optimization.Frontiers in Bioengineering and Biotechnology. 2022, 10: 865820.

（3）An adaptive low-illumination image enhancement algorithm based on weighted least squares optimization. Journal of Physics: Conference Series. IOP Publishing. 2022, 2181(1): 012011.

（4）Non-Uniform-Illumination Image Enhancement Algorithm Based on Retinex Theory. Applied Sciences. 2023, 13(17): 9535.

Review 3：

1. Aiming at the problem that the methods in related fields introduced in the introduction part need to be more novel, we have improved them in the revised draft by quoting the latest research methods and elaborating them in detail. These methods are as follows:

（1）Natural digital image mixed noise removal using regularization Perona–Malik model and pulse coupled neural networks.Soft Computing. 2023: 1-10. 

（2）Underwater image enhancement by attenuated color channel correction and detail preserved contrast enhancement. IEEE Journal of Oceanic Engineering 2022, 47(3): 718-735. 

（3）Natural image noise removal using nonlocal means and hidden Markov models in transform domain. The Visual Computer. 2018, 34: 1661-1675. 

（4）Underwater image enhancement via piecewise color correction and dual prior optimized contrast enhancement. IEEE Signal Processing Letters. 2023, 30: 229-233. 

（5）Additive Gaussian noise removal based on generative adversarial network model and semi-soft thresholding approach. Multimedia Tools and Applications.2023, 82(5): 7757-7777. 

2. We take the problem of experts pointing out the working block diagram seriously. We draw on the paper "Underwater Image Enhancement by Attenuated Color Channel Correction and Detail Preserved Contrast " in the revised version. In the method overview diagram in "Enhancement," the working block diagram is redrawn, and the corresponding visual effects and enhanced result diagram are added. Under the working block diagram in the introduction section, we describe the specific operation steps in three detailed parts to provide a clear understanding and reference. 

3. As for the parameters in Formula (6), we explain the necessary meaning of the parameters in the revised manuscript to ensure readers' correct understanding of the parameters. 

4. In addition, in view of the lack of comparison experiments with the latest method enhancement methods in the investigation, we introduce the four latest techniques in the revised manuscript for comparison experiments. Specific citations are as follows:

（1）Low-Light Image Enhancement Method Based on Retinex Theory by Improving Illumination Map. Applied Sciences. 2022, 12(10): 5257.

（2）Low-illumination image enhancement algorithm based on improved multi-scale Retinex and ABC algorithm optimization.Frontiers in Bioengineering and Biotechnology. 2022, 10: 865820.

（3）An adaptive low-illumination image enhancement algorithm based on weighted least squares optimization. Journal of Physics: Conference Series. IOP Publishing. 2022, 2181(1): 012011.

（4）Non-Uniform-Illumination Image Enhancement Algorithm Based on Retinex Theory. Applied Sciences. 2023, 13(17): 9535.

5. Given the problem of formula subscripts proposed by experts, we conducted a comprehensive investigation and correction in the revised draft to ensure that the subscripts of all formulas have been corrected from italics to positive. 

6. At the same time, given the clarity problem of the drawing pointed out by experts, we optimized the relevant pictures in the revised draft to improve the overall clarity and readability.

---

## [Decision Letter · Decision Letter 2]

15 Jan 2024

PONE-D-23-30473R2Low illumination fog noise image denoising method based on ACE-GPMPLOS ONE

Dear Dr. wang,

Thank you for submitting your manuscript to PLOS ONE. After careful consideration, we feel that it has merit but does not fully meet PLOS ONE’s publication criteria as it currently stands. Therefore, we invite you to submit a revised version of the manuscript that addresses the points raised during the review process.

We look forward to receiving your revised manuscript.

Kind regards,

Sen Xiang

Academic Editor

PLOS ONE

Journal Requirements:

**Additional Editor Comments:**

 1. Please cite paper published in 20232. Please improve the organization.

Reviewers' comments:

Reviewer's Responses to Questions

**Comments to the Author**

1. If the authors have adequately addressed your comments raised in a previous round of review and you feel that this manuscript is now acceptable for publication, you may indicate that here to bypass the “Comments to the Author” section, enter your conflict of interest statement in the “Confidential to Editor” section, and submit your "Accept" recommendation.

Reviewer #1: All comments have been addressed

Reviewer #2: (No Response)

Reviewer #3: All comments have been addressed

2. Is the manuscript technically sound, and do the data support the conclusions?

Reviewer #1: Yes

Reviewer #2: No

Reviewer #3: Yes

3. Has the statistical analysis been performed appropriately and rigorously? 

Reviewer #1: Yes

Reviewer #2: No

Reviewer #3: Yes

4. Have the authors made all data underlying the findings in their manuscript fully available?

Reviewer #1: Yes

Reviewer #2: No

Reviewer #3: Yes

5. Is the manuscript presented in an intelligible fashion and written in standard English?

Reviewer #1: Yes

Reviewer #2: No

Reviewer #3: Yes

6. Review Comments to the Author

Reviewer #1: (No Response)

Reviewer #2: The Recent manuscripts are not cited. The 2023 recent works need to be cited. Author should take proper care in organizing the manuscript.

Reviewer #3: The authors have well addressed all my concerns in the revision. The manuscript is acceptable for publication in its present form.

7. PLOS authors have the option to publish the peer review history of their article (what does this mean?). If published, this will include your full peer review and any attached files.

Reviewer #1: No

Reviewer #2: **Yes: **Malaya Kumar Nath

Reviewer #3: No

---

## [Author Response · Author response to Decision Letter 2]

9 Mar 2024

Thank you for taking time out of your busy schedule to read my manuscript and for your reply letter. For the questions and opinions of the experts, I will reply one by one.

Comments to the editors:

1.This paper re-cited the papers published in 2023-2024, and the updated literatures are as follows: 

Reference 8：Plutino A, Tarini M. Fast ACE (FACE): an error-bounded approximation of Automatic Color Equalization[J]. IEEE Transactions on Image Processing, 2023. 

Reference 9: Wang D, Liu Z, Gu X, et al. Feature extraction and segmentation of pavement distress using an improved hybrid task cascade network[J]. International Journal of Pavement Engineering, 2023, 24(1): 2266098. 

Reference 10: Ning G. Two-dimensional Otsu multi-threshold image segmentation based on hybrid whale optimization algorithm[J]. Multimedia Tools and Applications, 2023, 82(10): 15007-15026 

Reference 11：Du Y, Yuan H, Jia K, et al. Research on Threshold Segmentation Method of Two-Dimensional Otsu Image Based on Improved Sparrow Search Algorithm[J]. IEEE Access, 2023 

Literature 14: Raonic B, Molinaro R, De Ryck T, et al. Convolutional neural operators for robust and accurate learning of PDEs[J]. Advances in Neural Information Processing Systems, 2024, 36 

Reference 15: Zhang Q, Song X, Song S, et al. Finite-Time sliding mode control for singularly perturbed PDE systems[J]. Journal of the Franklin Institute, 2023, 360(2): 841-861 

Among them, references 8 and 9 are the latest research on Automatic Color Equalization algorithm (ACE), references 10 and 11 are the latest research on the maximum between-cluster variance method, and references 14 and 15 are the latest research and application on partial differential equations.

2.In view of the problems in the organization of the article, this article has been updated as follows:

(1)The experimental part has been adjusted, the dataset used in this article has been summarized, and the GitHub access address has been provided: https://github.com/WuyiLi1999/Low_illumination_fog_noise_images.git. 

(2)The original chapter of experimental environment, benchmark model experiment, model evaluation index and model parameter setting is integrated to form the chapter of "experimental setup". 

(3)In the chapter of experimental results and analysis, the introduction of the comparison model, the comparison of advantages and disadvantages are organized. At the same time, the position of the comparison result chart and the evaluation index data table are updated, which makes the article more intuitive.

Regarding the comments of the three reviewers: 

1.Whether to respond to the previous round of comments: Reviewer 2 did not receive any response to his comments, so we will keep the previous response below for review by reviewer 2

2.Answer the question "Is the manuscript technically sound, and does the data support the conclusions?" Question: In the experimental analysis stage, this paper analyzes multiple groups of visual effects and FOM merit evaluation indexes, and draws corresponding conclusions according to the data in the table. Time complexity analysis and ablation experiments are carried out to verify the proposed model.

3.Is the statistical analysis performed appropriately and rigorously? Problem: This paper uses 7 evaluation metrics to perform multiple group analysis and carries out time complexity analysis and visual analysis of processing time.

4.In response to the question "Did the authors make all data in their manuscript fully available?" Question: The experimental result data in this paper is based on the content in Section 3.2 Experimental setup. Please refer to the GitHub address for the calculation code of the data set and evaluation index data.

5.In response to the question "Is the manuscript presented in an understandable way and written in standard English? Problem: The author once again checked the spelling and grammar of the words in this article and corrected any problems found.

6.For the question of "reviewing comments to authors" : the comments to reviewer 1 have been submitted before. If reviewer 1 has not received the comments, the reviewer should reply again in this submission response; In response to the comments of reviewer 2, this paper re-cited the recently published literature. Please refer to the updated literature as follows:

Reference 8：Plutino A, Tarini M. Fast ACE (FACE): an error-bounded approximation of Automatic Color Equalization[J]. IEEE Transactions on Image Processing, 2023. 

Reference 9: Wang D, Liu Z, Gu X, et al. Feature extraction and segmentation of pavement distress using an improved hybrid task cascade network[J]. International Journal of Pavement Engineering, 2023, 24(1): 2266098. 

Reference 10: Ning G. Two-dimensional Otsu multi-threshold image segmentation based on hybrid whale optimization algorithm[J]. Multimedia Tools and Applications, 2023, 82(10): 15007-15026 

Reference 11：Du Y, Yuan H, Jia K, et al. Research on Threshold Segmentation Method of Two-Dimensional Otsu Image Based on Improved Sparrow Search Algorithm[J]. IEEE Access, 2023 

Literature 14: Raonic B, Molinaro R, De Ryck T, et al. Convolutional neural operators for robust and accurate learning of PDEs[J]. Advances in Neural Information Processing Systems, 2024, 36 

Reference 15: Zhang Q, Song X, Song S, et al. Finite-Time sliding mode control for singularly perturbed PDE systems[J]. Journal of the Franklin Institute, 2023, 360(2): 841-861 

Given the problem that Expert 1 and Expert 2 pointed out that they did not receive relevant responses in the last round of review, we at this moment make the following remarks: In the previous submission, the PDF called "Response to Reviewers" had pages 3 and 4 in response to Expert 1's review, page5 in response to expert 2's review, and page 6 in response to expert 3's review.Since the experts could not see the corresponding reply, I will attach the related reply content from the last time for your reference in this reply. First, the response to the previous round of expert 1's opinion is as follows:

First, the response to the previous round of expert 1's opinion is as follows:

一、Abstract： 

1. The updated abstract reflects this paper's image reconstruction method steps. Firstly, an 

automatic color equalization algorithm(ACE) is used to improve the contrast of the low-light 

image containing noise. Secondly, the maximum between-cluster variance method(OTSU) was 

used for threshold segmentation. Then, according to the segmentation threshold, the image was 

enhanced by gradient using the enhancement function. Finally, the P-M model was used to denoise 

and enhance the gradient image. 

2. The method proposed in this paper is mainly elaborated at the beginning of the third sentence 

of the abstract. 

3. The title is updated to ： Low illumination fog noise image denoising method based on 

ACE-GPM 

4. The comparison results between the proposed method and other methods are reflected in the 

abstract. Compared with other denoising methods, the ACE-GPM method proposed in this paper 

improves image contrast and removes noise while effectively retaining image details and texture 

information. The information entropy value is increased by 0.42 on average. In addition, the 

proposed method requires fewer computational resources while maintaining image quality. 

二、Introduction： 

In the introduction of the revised version, this paper first expounds on the common types of noise 

in digital images and the reasons for their generation. Subsequently, we explain the denoising 

methods proposed in the literature and their advantages and disadvantages. Finally, we show the 

proposed method's working block diagram and method steps so readers can better understand our 

method. 

三、Method： 

1. In the ninth paragraph of the introduction, this paper elaborates on the anisotropic P-M 

diffusion equation model and its advantages and disadvantages in image denoising and 

enhancement. This model has a wide range of application values in image processing, but due to 

its limitations, some areas also need improvement. 

2. In the research method section of this paper, part D elaborates in-depth, and how to apply the 

OTSU method for gradient enhancement is introduced in detail. At the same time, we also clarify 

how the method ADAPTS to the P-M model and its embodiment of the performance improvement 

of the P-M model. 

3. At the end of the introduction, this paper shows a simple working block diagram of the 

proposed method and elaborates each step in detail. 

四、Experiment： 

1. In the revised manuscript's experimental part, this paper adds two sets of visual contrast and 

FOM merit evaluation to verify the effectiveness and reliability of the proposed method 

comprehensively. These additional experimental groups are intended to provide more exhaustive 

empirical support for our approach. 

2. The revised manuscript details the hardware and software specifications in Section A of the 

experiment. The introduction section clearly describes the working steps of the proposed method.Each part of the research method constitutes the processing flow of the method in turn. 

3. This paper studies the noise reduction problem of low-light fog images. We propose an 

effective denoising method for fog noise in low-light fog images. 

4. This paper selects low-light fog noise images taken under natural conditions, and the 

comparison results between visual benchmark images and images processed by the method are 

provided in Section D of the experiment. 

5. In Table 2 of the original paper, we detail the objective index evaluation of the images 

processed by various methods, including indicators such as average brightness, standard deviation, 

information entropy, peak signal-to-noise ratio (PSNR), and similarity index (SSIM). Because 

these evaluation metrics are not explained in detail in the original paper, it may have brought you 

questions. Therefore, in Section B of the experimental section of the revised manuscript, we 

present the FOM merit evaluation index and its calculation formula in this paper. In Tables 3, 4, 

and 5 in Section D, we detail the FOM merit evaluation index data of the processed images by 

each denoising algorithm. 

6. According to the data in Table 2 of the original paper, the running time of various methods for 

processing waterfall images may need to be more intuitive to show their effects. Therefore, after 

expert advice, this paper shows the line chart of running time for images with different resolutions 

in FIG. 7, part E of the experiment of the revised manuscript. This gives the reader a clearer 

picture of how different processing methods differ in running time and helps the reader evaluate 

the performance of each method more accurately. 

7. The revised version of experiment section E shows the evaluation data of each part of the 

proposed method and the total time complexity. 

8. In the previous version, the introduction of OTSU's algorithm may have some things that could 

be improved. Based on the expert feedback, this paper gives a more comprehensive description of 

the OTSU threshold method. It explains in detail how it is applied to the P-M model in Section B 

of the revised version of research methods. 

9. With revisions and updates, the references and conclusions in this paper have been refined. In 

the references, we included some crucial articles published recently to ensure the timeliness and 

relevance of this paper. At the same time, in the conclusion section, we point out some limitations 

of the proposed method. We still need to adaptively select the P-M model's critical parameters for 

different pixel and definition images.

Secondly, the response to Expert 3's last round of comments is as follows:

1.Abstract：We have revised and updated the paper's abstract according to the expert advice. Firstly, this paper leads to the proposed method for the problems of the existing P-M model. Secondly, the implementation of the method proposed in this paper is briefly described. Finally, the experimental results comparing the proposed method with other denoising methods in the literature are described to prove the effectiveness of the proposed method.

2.Introduction：In the revised manuscript's introduction section, we describe this paper's workflow in detail and clearly explain the tasks of each stage. In this way, we can more accurately communicate our work's content, goals, and importance so that readers can better understand our work.

3.References Cited: After revision, all references cited in this paper have been updated to ensure timeliness and relevance.

4.Format and formula: After carefully reviewing and revising the first draft and referring to the reference articles provided by experts, we have revised the format and formula to ensure that the final revised draft meets the rigorous, stable, rational, and official requirements in form and content.

5.Flowchart of the paper framework: At the end of the introduction section of the revised manuscript, we attach a flowchart of the work of this paper, which details the tasks and objectives of each stage.

6.Image clarity: In the revised version, to more fully demonstrate the effectiveness of the proposed method, we conduct more comparative experiments and improve the image's contrast.

7.Ablation Experiment: In the revised version, the ablation experiment is added to Part F of an experiment to evaluate the influence of each part of the method and the combination of each part on the P-M model to reflect the effectiveness of the improved strategy in this paper.

Finally, a reply to the opinions of the experts in this round:

Review 1：

一、Comments on the Introduction: 

1. According to your professional suggestions, we have carefully revised the middle part of the third page of the revised manuscript to focus on the core contributions of this paper. In addition, we have redrawn the system block diagram and refer to the paper "Underwater Image Enhancement by Attenuated Color Channel Correction and Detail Preserved Contrast. The overview diagram of the method in Enhancement "draws a block diagram of the working of the proposed method. The work block diagram shows the graphical information of each step in detail to help readers understand the research content of this paper more intuitively. 

2. In the introduction section, we discuss five representative noise-denoising algorithms in-depth. The following is the literature corresponding to these denoising algorithms:

1.Paper 1：Natural digital image mixed noise removal using regularization Perona–Malik model and pulse coupled neural networks.Soft Computing. 2023: 1-10. 

2. Paper 2：Underwater image enhancement by attenuated color channel correction and detail preserved contrast enhancement. IEEE Journal of Oceanic Engineering 2022, 47(3): 718-735. 

3. Paper 3：Natural image noise removal using nonlocal means and hidden Markov models in transform domain. The Visual Computer. 2018, 34: 1661-1675. 

4. Paper 4：Underwater image enhancement via piecewise color correction and dual prior optimized contrast enhancement. IEEE Signal Processing Letters. 2023, 30: 229-233. 

5. Paper 5：Additive Gaussian noise removal based on generative adversarial network model and semi-soft thresholding approach. Multimedia Tools and Applications.2023, 82(5): 7757-7777. 

二、Comments on the experiment:

1.According to your advice from experts, to objectively evaluate the quality of test images, we have added evaluation indicators such as peak mean square Error (RMSE), Square Absolute Error (AME), and mean square error of Structural Similarity (SDME). These metrics help to measure image quality more comprehensively, and their specific mathematical formulas have been detailed in Experiment Section B. With these improvements, we can evaluate the performance of various methods in terms of image quality more accurately. 

2.To verify the effectiveness of the proposed enhancement method, this paper compares the latest representative image enhancement methods and briefly describes their algorithm process, advantages, and disadvantages. The following is a list of literature corresponding to enhancement methods:

---

## [Decision Letter · Decision Letter 3]

26 Mar 2024

PONE-D-23-30473R3Low illumination fog noise image denoising method based on ACE-GPMPLOS ONE

Dear Dr. wang,

Thank you for submitting your manuscript to PLOS ONE. After careful consideration, we feel that it has merit but does not fully meet PLOS ONE’s publication criteria as it currently stands. Therefore, we invite you to submit a revised version of the manuscript that addresses the points raised during the review process.

 **Please further revise the paper as Reviewer 2# suggested before it can be published. **

We look forward to receiving your revised manuscript.

Kind regards,

Sen Xiang

Academic Editor

PLOS ONE

Journal Requirements:

Additional Editor Comments:

**Although two of three reviewers suggest acceptance, one reviewer suggest another round of revision. I would like to suggest the author revise the manuscript again before it can be accepted.**

Reviewers' comments:

Reviewer's Responses to Questions

**Comments to the Author**

1. If the authors have adequately addressed your comments raised in a previous round of review and you feel that this manuscript is now acceptable for publication, you may indicate that here to bypass the “Comments to the Author” section, enter your conflict of interest statement in the “Confidential to Editor” section, and submit your "Accept" recommendation.

Reviewer #2: All comments have been addressed

2. Is the manuscript technically sound, and do the data support the conclusions?

Reviewer #2: Yes

3. Has the statistical analysis been performed appropriately and rigorously? 

Reviewer #2: Yes

4. Have the authors made all data underlying the findings in their manuscript fully available?

Reviewer #2: Yes

5. Is the manuscript presented in an intelligible fashion and written in standard English?

Reviewer #2: Yes

6. Review Comments to the Author

Reviewer #2: The reviewers comments are not properly incorporated in the manuscript.

The manuscript need to be thoroughly checked prior to submission

7. PLOS authors have the option to publish the peer review history of their article (what does this mean?). If published, this will include your full peer review and any attached files.

Reviewer #2: No

---

## [Author Response · Author response to Decision Letter 3]

28 Mar 2024

Thank you for taking time out of your busy schedule to read my manuscript and for your reply letter. For the questions and opinions of the experts, I will reply one by one.

1.Title : This manuscript has been carefully revised, and the title has been precisely adjusted in response to the problem raised by the reviewers. The revised title is "Low illumination fog noise image denoising method based on ACE-GPM" to more accurately reflect this paper's research content and focus.

2.Abstract： Based on the reviewers' valuable comments, the “abstract ” part has been carefully optimized. In the abstract, we first point out the existing problems, clarify the main research methods adopted, and give a brief overview of the experimental results as the concluding part of the abstract. Such modifications help readers quickly understand this paper's core content and research results.

3.Introduction： After careful consideration of the reviewers' valuable comments on the introduction section, this manuscript has revised the introduction section accordingly. The revised content highlights the innovation of this paper even more and clarifies the overall architecture of the research methodology. We adopted the reviewers' suggestions to make this paper more rigorous, sedate, rational, and in line with the requirements of the official language style.

4.Formula：After careful consideration of the valuable suggestions of the reviewers on the formula, this revised version of the manuscript comprehensively reviewed and revised the formula involved and elaborated and explained the various parameters in the formula to ensure the logic and rigor of the paper.

5.Framework flow chart: After the reviewer pointed out the lack of a paper framework flow chart in the original manuscript, the revised version of this manuscript actively adopted the reviewer's suggestions. It referred to the literature resources recommended by the reviewer. After careful modification and improvement, we draw a detailed framework flow chart of the research method in this paper and give a comprehensive and in-depth explanation of the various components within the framework. Through these improvements, we expect to significantly improve the readability and intelligibility of the paper and provide readers with a more transparent and intuitive display of research ideas and methodologies. Refer to the last section of the introduction in the paper for a detailed framework flowchart and illustration.

6.Low resolution: After careful consideration and review, the revised version of this manuscript has been thoroughly checked and addressed, given the low resolution of the images raised by the reviewers. We have made the necessary optimizations and upgrades to the pictures to ensure their clarity and resolution reach a higher standard, thus more accurately conveying the research content and results. These improvements will contribute to the overall quality and readability of the manuscript.

7.Ablation experiments: After careful consideration of the valuable comments of the reviewers, we have supplemented the revised manuscript with ablation experiments. Specifically, we conduct relevant Settings in Section 3.5 of the paper and verify the effectiveness of the proposed improved scheme through an exhaustive analysis. This addition is intended to fully demonstrate the depth and breadth of the research while also providing readers with clearer and more robust evidence support.

8.Spelling and grammatical: In response to the reviewers' comments on spelling and grammatical errors in this manuscript, we will review the paragraphs one by one and accurately revise the sentences with grammatical errors to ensure the accuracy and fluency of the text.

9.Conclusions: After carefully considering the reviewers' comments on this paper's conclusions, we have decided to revise and improve the findings in the manuscript. The revised conclusion not only elaborates on the workflow and performance of the proposed method but also deeply analyzes the limitations of the proposed method and important directions for future research. Such revisions aim to improve the scientific and rigorous nature of the study and provide more accurate and comprehensive references for researchers in related fields.

10.charts and graphs: After careful study of the reviewers' feedback on the charts and graphs of the manuscript, we have decided to focus on the presentation of charts and tables in this revision and to improve their readability and accuracy through appropriate magnification and optimization measures.

11.Reference: Given the problem of inconsistent reference format raised by the reviewers, we have carried out a comprehensive revision of the reference section in the original manuscript. This revision aims to ensure that all cited documents follow a uniform formatting standard, thus improving the academic normativity and readability of the paper. We thank the reviewers for their valuable comments. We will continue to work hard to improve the quality of the paper.

12.Introduction: Based on the suggestions the reviewers put forward, this revised manuscript's introduction first summarizes the common noise types and their corresponding denoising algorithms. It makes it clear that the main research object of this paper is the low-light fog noise image. Through carefully designed experiments, we verify that the proposed method can effectively preserve the detailed texture information of the image while removing the fog noise in the picture, ensuring that the image quality is comprehensively improved.

13.Time complexity: Given the problem raised by the reviewers that the time complexity of the method needed to be evaluated in the manuscript, this revised manuscript has added a detailed analysis of the time complexity in Section 3.4.

14.OTSU's threshold: In response to the reviewer's feedback on the need for in-depth elaboration of OTSU's threshold, this revised manuscript elaborates on the specific steps of OTSU's algorithm in Section 2.2. It explores in depth the performance improvement that can be achieved by the algorithm in this study. We are committed to providing precise and detailed instructions to ensure readers fully understand the specific application and impact of OTSU's threshold in this study.

15.FOM merit evaluation index: In response to the reviewer's request for a more comprehensive evaluation of the method's performance in this paper, the revised manuscript has added multiple reliable performance evaluation indicators based on the original experimental setup. It includes peak mean square error (RMSE), Mean absolute error (AME), and mean square error of structural similarity (SDME). These new evaluation indexes will form a more complete evaluation system to reflect the method's performance in practical applications accurately.

---

## [Editor Report · Decision Letter 4]

5 Apr 2024

Low illumination fog noise image denoising method based on ACE-GPM

PONE-D-23-30473R4

Dear Dr. wang,

We’re pleased to inform you that your manuscript has been judged scientifically suitable for publication and will be formally accepted for publication once it meets all outstanding technical requirements.

Kind regards,

Sen Xiang

Academic Editor

PLOS ONE